# A high-throughput screen for TMPRSS2 expression identifies FDA-approved compounds that can limit SARS-CoV-2 entry

Yanwen Chen [1,2,7], Travis B. Lear [1,3,4,7], John W. Evankovich [1,3,7], Mads B. Larsen [1], Bo Lin [1], Irene Alfaras [1], Jason R. Kennerdell[1], Laura Salminen[1], Daniel P. Camarco[1], Karina C. Lockwood[1], Ferhan Tuncer[1], Jie Liu[1], Michael M. Myerburg[3], John F. McDyer[3], Yuan Liu [1,3,5 ✉], Toren Finkel [1,4,6 ✉] & Bill B. Chen [1,3,4 ✉]

SARS-CoV-2 (2019-nCoV) is the pathogenic coronavirus responsible for the global pandemic of COVID-19 disease. The Spike (S) protein of SARS-CoV-2 attaches to host lung epithelial cells through the cell surface receptor ACE2, a process dependent on host proteases including TMPRSS2. Here, we identify small molecules that reduce surface expression of TMPRSS2 using a library of 2,560 FDA-approved or current clinical trial compounds. We identify homoharringtonine and halofuginone as the most attractive agents, reducing endogenous TMPRSS2 expression at sub-micromolar concentrations. These effects appear to be mediated by a drug-induced alteration in TMPRSS2 protein stability. We further demonstrate that halofuginone modulates TMPRSS2 levels through proteasomal-mediated degradation that involves the E3 ubiquitin ligase component DDB1- and CUL4-associated factor 1 (DCAF1). Finally, cells exposed to homoharringtonine and halofuginone, at concentrations of drug known to be achievable in human plasma, demonstrate marked resistance to SARS-CoV-2 infection in both live and pseudoviral in vitro models. Given the safety and pharmacokinetic data already available for the compounds identified in our screen, these results should help expedite the rational design of human clinical trials designed to combat active COVID-19 infection.

---

[1] Aging Institute, University of Pittsburgh/UPMC, Pittsburgh, PA, USA. [2] Department of Gastroenterology, Ruijin Hospital, Shanghai Jiaotong University School of Medicine, Shanghai, China. [3] Department of Medicine, Acute Lung Injury Center of Excellence, University of Pittsburgh, Pittsburgh, PA, USA. [4] Vascular Medicine Institute, University of Pittsburgh, Pittsburgh, PA, USA. [5] McGowan Institute for Regenerative Medicine, University of Pittsburgh, Pittsburgh, PA, USA. [6] Department of Medicine, Division of Cardiology, University of Pittsburgh, Pittsburgh, PA, USA. [7]These authors contributed equally: Yanwen Chen, Travis B. Lear, John W. Evankovich. ✉email: liuy13@upmc.edu; finkelt@pitt.edu; chenb@upmc.edu

SARS-CoV-2 is a coronavirus first described in Wuhan, China that shares many similarities with other pathogenic beta coronaviruses, including the SARS-CoV virus and the MERS coronavirus[1]. After emergence of SARS-CoV, several groups identified the molecular and cellular pathways through which this virus attaches, enters, and replicates in host respiratory epithelial cells[2–4]. To mediate cell entry, the SARS-CoV viral Spike (S) glycoprotein is recognized by the extracellular receptor angiotensin converting enzyme 2 (ACE2) on host respiratory epithelial cells[5,6]. A second critical process, S protein priming, is essential to complete viral entry and spread[7]. S protein priming is a process wherein the spike protein is cleaved by host proteases. Two separate classes of proteases were shown to prime SARS-CoV S protein—endo-lysosomal proteases in the cathepsin family, and the plasma membrane associated protease, TMPRSS2[3,8]. The viral entry mechanisms elucidated for SARS-CoV infection appear to be shared by SARS-CoV-2[9,10]. Additionally, SARS-CoV-2 uniquely possesses a "furin-like" cleavage site and is subject to processing during viral packaging by the intracellular proprotein convertase furin[11–14].

Host-directed efforts to limit SARS-CoV-2 infection could logically entail strategies to manipulate either ACE2 or the S protein priming step. However, in animal models, reducing ACE2 activity appears to increase lung injury in response to SARS-CoV infection or sepsis[6,15]. In contrast, mice lacking TMPRSS2 exhibit

no discernable basal phenotype[16], but demonstrate protection from acute SARS-CoV infection[3]. These observations led us, and others[9,17], to hypothesize that reducing or inhibiting TMPRSS2 may be an attractive strategy to mitigate SARS-CoV-2 pathogenicity. Given SARS-CoV-2's rapid spread, coupled with the high burden of acute respiratory failure and death, there is an urgent need for therapies. We developed an assay that measures protein abundance of TMPRSS2 and executed a screen of 2560 FDA-approved or current clinical trial compounds to determine their effect on TMPRSS2 abundance. Based on our screening strategy and already published pharmacokinetic and toxicology profiles, we identified compounds halofuginone and homoharringtonine as agents that reduce cell surface expression of TMPRSS2. Both halofuginone and homoharringtonine reduced entry of pseudotyped SARS-CoV-2 in TMPRSS2-expressing lung epithelial cells, and significantly reduced Calu-3 infection with authentic SARS-CoV-2. Given our screening approach to identify compounds that affect protein abundance, we also characterized how TMPRSS2 is targeted for E3 ligase driven ubiquitination and proteasomal degradation. We identify the E3 ligase component DDB1- and CUL4-associated factor 1 (DCAF1) as critical for regulating TMPRSS2 stability and show the effect of halofuginone to reduce TMPRSS2 abundance is DCAF1-dependent.

## Results

In order to assess agents that might alter TMPRSS2 expression, we first engineered full-length human TMPRSS2 to express a C-terminal 11 amino acid tag (HiBiT), which produces a bioluminescent signal when combined with a complementary protein (LgBiT) and a furimazine substrate[18]. The HiBiT-tagged TMPRSS2 construct was expressed in human bronchial epithelial cells (Beas-2B), which have been utilized as model cell lines for several infectious airway diseases[19,20]. We utilized two detection reagents, which allowed us to detect either plasma membrane-associated TMPRSS2 with a non-lytic nano-luciferase (Nano-Luc) reagent, or total TMPRSS2 expression using a lytic luciferase agent (Fig. 1). Elements of the screen were optimized and miniaturized to a 384-well format. We aimed to discover compounds that reduced TMPRSS2 abundance; thus, we calculated the Z′ score of cells treated with vehicle control (Veh) or cycloheximide, a global protein translation inhibitor that reduced TMPRSS2-HiBiT signal. The Z′ score was 0.38, and thus suitable for high-throughput applications (Fig. S1).

Next, we screened a chemical library (FDA-approved and current clinical trial drugs) consisting of 2560 compounds (Selleck, L1300) for their capacity to decrease TMPRSS2-HiBiT abundance (Fig. 2A, please see supplemental source data). We assayed the compounds effect on both the extracellular plasma membrane associated and total TMPRSS2-HiBiT signal using the non-lytic and lytic luciferase reagent. At a compound concentration of 10 μM, this screen identified 100 drugs that produced a >60% reduction in extracellular TMPRSS2-HiBiT signal (Fig. 2A). A number of compounds reduced both extracellular and total TMPRSS2 levels (Fig. 2B: intersection of purple and pink rectangles). In contrast, the agent nafamostat, a serine protease inhibitor, produced the opposite effect, as treatment with this agent potently increased TMPRSS2 abundance (Fig. 2B). This compound is known to block TMPRSS2 enzymatic activity and thereby serves as a potential strategy to limit SARS-CoV and MERS acute infection of cells[21]. A related compound, camostat, is also known to enzymatically inhibit TMPRSS2, and was recently shown to block acute SARS-CoV-2 infection[9], prompting the initiation of a clinical trial for this agent in COVID-19 patients (ClinicalTrials.gov Identifier: NCT04321096). However, our results suggest there might be a feedback mechanism through

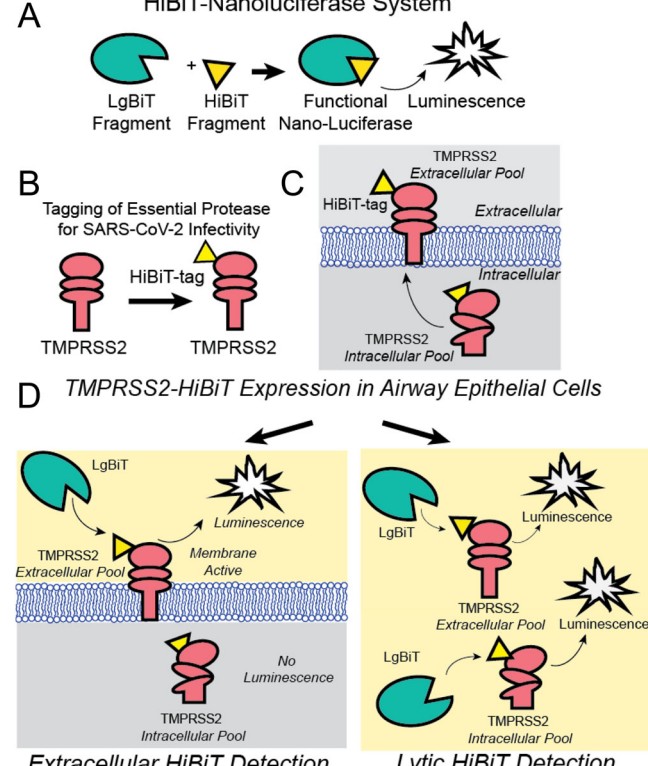

**Fig. 1 Schematic of TMPRSS2-HiBiT detection for high-throughput screen. A** The split nano-luciferase components LgBiT and HiBiT can interact to form a functional enzyme that generates luminescence. **B** Human TMPRSS2 cDNA was C-terminally tagged with a HiBiT sequence on a domain that is extracellular when TMPRSS2 is present in the plasma membrane. **C** The TMPRSS2-HiBiT construct was expressed in human airway cells where it exists in an intracellular pool and a plasma membrane-associated pool. **D** Non-lytic extracellular HiBiT detection results in LgBiT-HiBiT complementation solely with the pool of plasma membrane localized TMPRSS2. Following extracellular HiBiT detection, cells are lysed and the total TMPRSS2- HiBiT is then quantified.

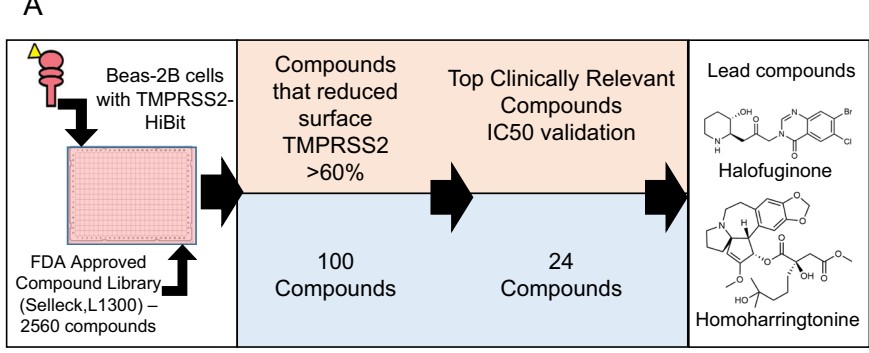

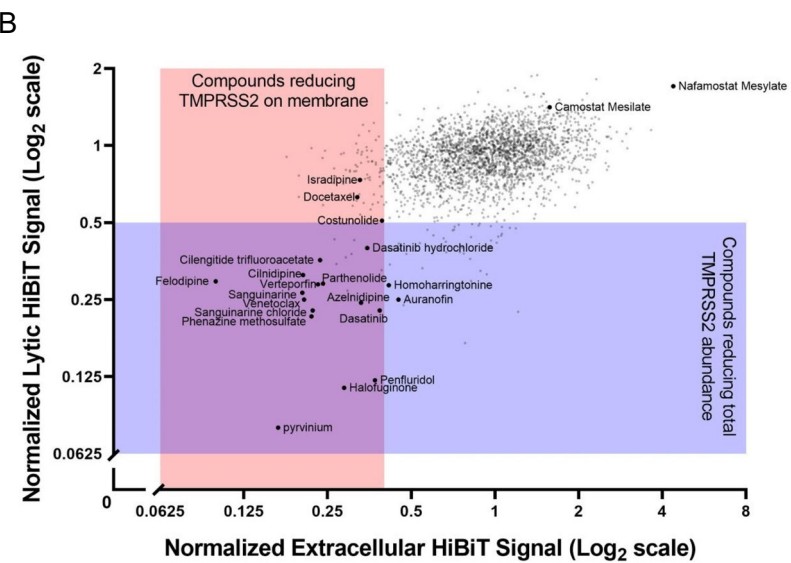

**Fig. 2 Screening of airway cells with a FDA-approved or clinically active compound library for agents that reduce TMPRSS2-HiBiT levels. A** Screening Assay workflow. Initial screening results yielded 100 compounds that decreased surface TMPRSS2 levels (by extracellular HiBiT detection) by 60%. From this, 24 compounds were selected for their clinical relevance and therapeutic potential and validated by IC50 assays. Of these, the most potent hits were halofuginone and homoharringtonine. **B** Scatterplot of hit compounds from both extracellular HiBiT and lytic HiBiT detection screens. Compounds that reduce membrane TMPRSS2-HiBiT signal (non-lytic extracellular HiBiT detection) are shown in pink, compounds that reduce total TMPRSS2-HiBiT signal (lytic HiBiT detection) are shown in purple. Some individual compounds are specified.

which TMPRSS2 enzymatic activity regulates overall levels of the protease. As such, these observations serve as a potentially cautionary note for using TMPRSS2 enzymatic inhibitors as a means to reduce coronavirus infections[9,21].

We carefully examined the top 100 compounds and focused on agents that were orally administered and had overall favorable clinical profiles. From this analysis, 24 compounds emerged (Fig. 2A). These compounds were cherry-picked and tested extensively in dose-course studies for their effect on the extracellular and intracellular TMPRSS2-HiBiT signal in airway epithelial cells (Fig. 3, Fig. S2 and Fig. S3). An assessment of the compound's toxicity, as measured by the drug's effect on cell number (using CellTiter-Glo 2.0) was also determined, so as to give an estimate of each compounds' potential therapeutic index (Fig. 3 and Fig. S2). As noted, many agents, including homoharringtonine (approved for chronic myeloid leukemia), halofuginone (in clinical trials for scleroderma) and cilnidipine (a calcium-channel blocker, antihypertensive agent approved in Asia and in some European countries), were effective in reducing extracellular TMPRSS2 expression at sub-micromolar concentrations. In general, these agents were also effective in reducing total TMPRSS2 as well, although for certain agents (e.g., cilnidipine) this required a substantially higher concentration of

drug. As an orthogonal approach to validate the effect of selected compounds on TMPRSS2 abundance, TMPRSS2-HiBiT protein expression was measured by HiBiT-blotting following compound treatment in mouse lung epithelial cells (MLE12) (Fig. S3).

The clinical utility of any agent identified ultimately depends on the ability of the potential therapeutic compound to reach presumed therapeutic levels in patients. As such, we compared the measured IC50 from our screen to the known pharmacokinetic properties of the identified agents. While we would ideally have preferred knowing the concentration of each drug in the presumptive target organ (e.g., lung), such information is not currently publicly available for many of these agents. As such, as an approximation, we leveraged the known concentration of each of these compounds in human plasma. From this exercise, homoharringtonine and halofuginone emerged as the candidates most likely to be clinically effective (Table 1).

We next sought to confirm that the reduction in the Nano-Luciferase signal observed in our screen by these two agents was not cell-type specific, dependent on the HiBiT tag, or a transcriptional effect of these compounds on the heterologous CMV promoter used to drive expression of our HiBiT-tagged TMPRSS2 construct. Using the same CMV promoter, we transiently expressed either a V5-tagged TMPRSS2 or a V5-tagged LacZ

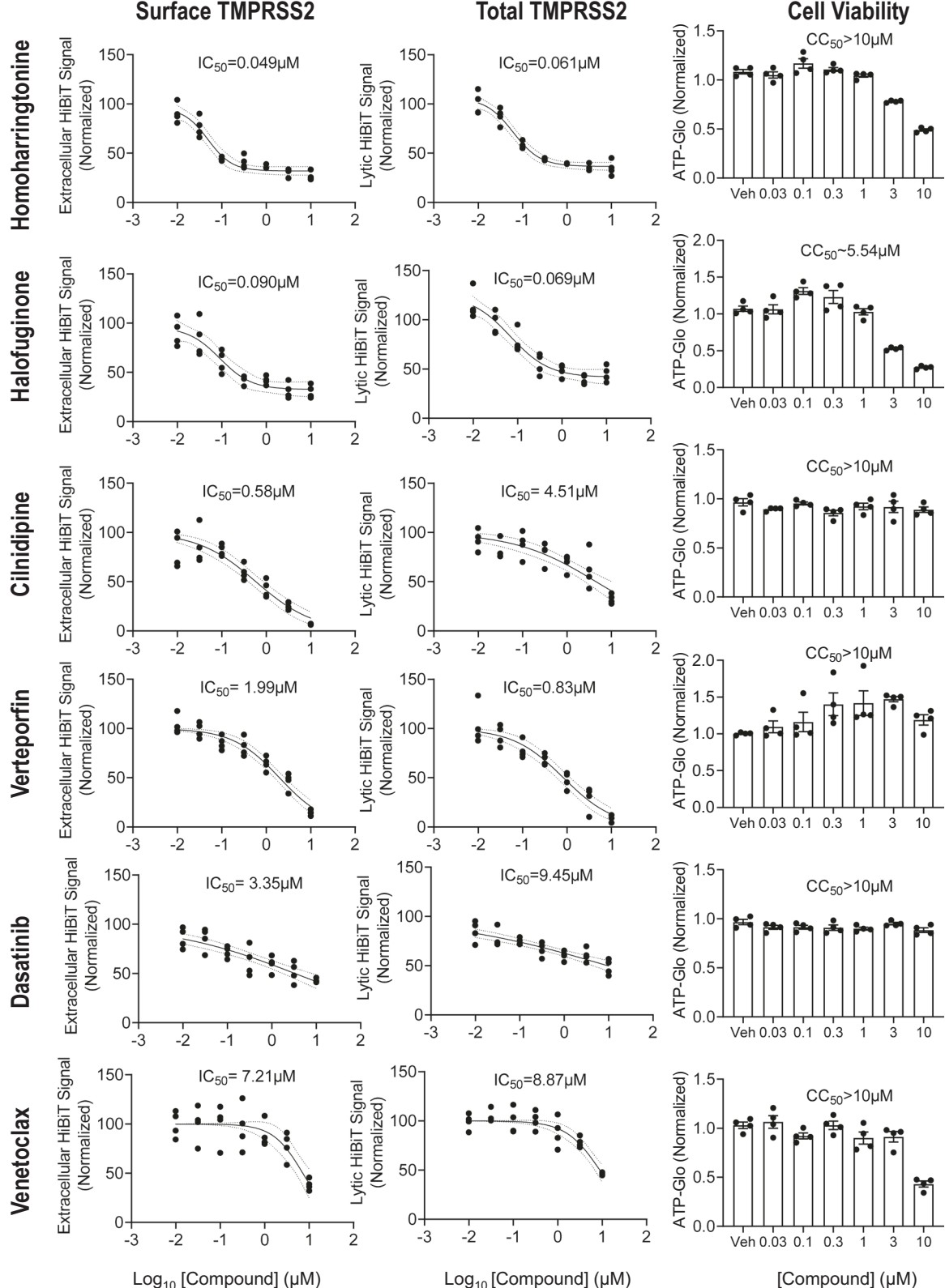

**Fig. 3 Determination of the potency for a subset of identified compounds.** A selection of six of the most promising drugs were assessed for their activity (IC50) to inhibit TMPRSS2 expression extracellularly (first column) or to inhibit total TMPRSS2 (middle column) in TMPRSS2-HiBiT-BEAS-2B cells. An assessment of cellular toxicity for each compound (CellTiter-Glo) is shown in the last column. The remaining activity profiles for other identified agents are found in the supplementary figures. Data are mean ± SEM ($n = 4$ biologically independent samples). Data are plotted along with a sigmoidal model fit, with dotted lines representing the 95% confidence interval. Estimated values of IC50 and CC50 (cytotoxicity concentration 50%) are shown.

**Table 1 Identified agents' current clinical indications, human pharmacokinetic properties[64-68] and estimated IC50 for inhibiting viral entry into Caco-2 cells.**

| Drug | Disease | Regulatory status | Dose | Route (frequency) | Cmax | In vitro IC50 (viral entry) | Reference |
|------|---------|-------------------|------|-------------------|------|------------------------------|-----------|
| Homoharringtonine (omacetaxine) | Chronic myeloid leukemia (CML) | FDA approved | 1.25 mg/m² | SC (BID) | 55 nM | ~30 nM (Fig. 7D) | 64 |
| Halofuginone | Scleroderma | Phase 1/2 | 3.5 mg/day | Oral | 7 nM | ~30 nM (Fig. 7E) | 65 |
| Cilnidipine | Hypertension | FDA approved | 10 mg | Oral (QD) | 18.1 nM | ~3 µM (Fig. S7C) | 66 |
| Dasatinib | Chronic myelogenous leukemia (CML) and acute lymphoblastic leukemia (ALL) | FDA approved | 140 mg | Oral (QD) | 0.307 µM | >10 µM (Fig. S7D) | 67 |
| Venetoclax | Chronic lymphocytic leukemia (CLL) or small lymphocytic lymphoma (SLL) | FDA approved | 400 mg | Oral (QD) | 1.27 µM | >10 µM (Fig. S7D) | 68 |

protein in the mouse respiratory epithelial cell line MLE-12. Both agents selectively reduced TMPRSS2 expression in cells (Fig. 4A). We then asked whether these compounds were effective in reducing endogenous TMPRSS2 expression. We chose the intestinal epithelial cell line Caco-2, since these cells are known to express high levels of TMPRSS2 and to be permissive for both SARS-CoV and SARS-CoV-2 infection[9,22]. Moreover, evidence suggests that the human intestinal tract may be an alternative entry point for coronaviruses[23]. Treatment of Caco-2 cells for 18 h with either homoharringtonine or halofuginone resulted in a marked reduction in endogenous TMPRSS2 protein expression (Fig. 4b, c). As expected, the observed decrease in TMPRSS2 protein expression was not a consequence of a drug-induced reduction in *TMPRSS2* transcription, which if anything, modestly increased with compound treatment (Fig. S3b). We also noted qualitatively similar decreases in TMPRSS2 protein levels in homoharringtonine or halofuginone-treated Calu-3 cells, a human lung cancer cell line that is also permissive to SARS-CoV-2 infection[9] (Fig. S4a, b). A time course demonstrated that the effects of drug treatment were relatively rapid, with a substantial reduction in TMPRSS2 expression evident within three hours (Fig. 4d, Fig. S4c). Similarly, levels of TMPRSS2 returned close to baseline roughly six hours after drug removal (Fig. 4e, Fig. S4d). Finally, these compounds showed specificity, as other membrane proteins such as E-cadherin, and Ace2 were unaffected by compound treatment (Fig. S5A, B).

While homoharringtonine is a relatively toxic agent with an adverse side effect profile[24,25], halofuginone has a very mild toxicity profile and is generally well tolerated[26,27]. Our primary screen identified compounds that modified stably expressed TMPRSS2-HiBiT abundance. While we focused on agents that reduced TMPRSS2-HiBiT, we also observed a number of agents that increased TMPRSS2-HiBiT signal, including several compounds that are proteasomal inhibitors. Thus, given our assay design and observed effects on TMPRSS2 abundance, we hypothesized that effects on TMPRSS2 abundance by halofuginone may be mediated in part by modifications in TMPRSS2 proteasomal-mediated disposal. In Caco-2 cells, the effect of halofuginone on TMPRSS2 protein levels was largely abrogated in the presence of the proteasomal inhibitor carfilzomib, but not the lysosomal inhibitor Bafilomycin A1 (Fig. 5A). Further, the NEDD8 E1 Activating Enzyme inhibitor MLN4924 had a similar effect to carfilzomib (Fig. S5c). This suggests that TMPRSS2 likely undergoes clearance through the ubiquitin-proteasomal system (UPS). The factors regulating TMPRSS2 degradation have not been described; thus, to identify potential elements of the UPS regulating TMPRSS2 abundance, we employed a high-throughput screen consisting of siRNAs targeting ~800 components of the UPS, including ubiquitin, proteasome subunits, E1, E2, E3, deubiquitinases (DUBs), and E3 ligases. As expected, knockdown of ubiquitin or proteasome subunits increased TMPRSS2 HiBiT signal robustly, consistent with our observations that the protein is subject to UPS-mediated turnover (Fig. 5B, C). While this screen showed knockdown of several of the proximal, non-specific elements of the UPS (E1, E2 enzymes) increased TMPRSS2, we also observed that knockdown of the E3 ligase subunit DCAF1 increased TMPRSS2 abundance (red dot) (Fig. 5B, C). E3 ligases are the most specific proteins in the UPS system, and are responsible for linking substrate proteins (TMPRSS2) to ubiquitin-transferring machinery for eventual disposal in the proteasome. To validate whether DCAF1 mediated TMPRSS2 proteasomal degradation, we overexpressed DCAF1 and observed decreased TMPRSS2-V5 protein in total cell lysate, and also increased poly-ubiquitination of TMPRSS2, suggesting a role of this E3 component in TMPRSS2 ubiquitination and half-life (Fig. 5D). Overexpression of DCAF1 also dose-dependently

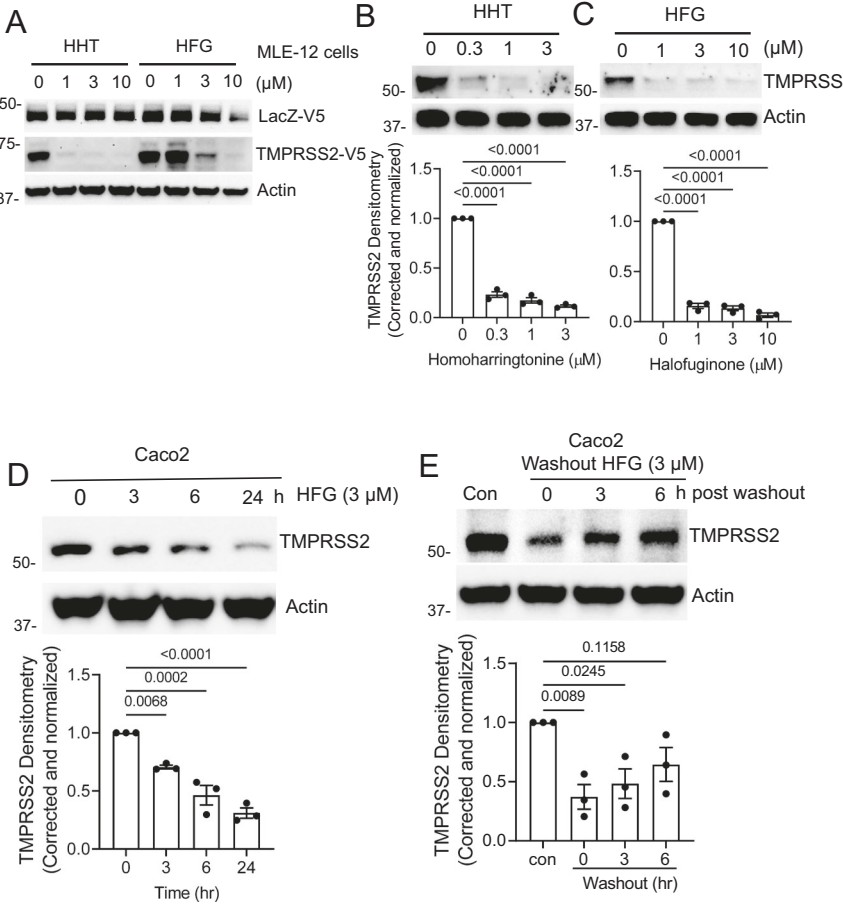

**Fig. 4 Homoharringtonine and halofuginone potently reduce TMPRSS2 protein levels. A** Immunoblot data from MLE-12 cells co-expressing LacZ-V5 and TMPRSS2-V5 treated with homoharringtonine (HHT) or halofuginone (HFG) at the indicated concentrations for 18 h. **B, C** Immunoblot analysis of endogenous TMPRSS2 protein level in Caco-2 cells treated for 18 h with HHT (**B**) or HFG (**C**). TMPRSS2 densitometry is shown, data are mean ± SEM ($n = 3$ biologically independent samples). **D** Time-course treatment of HFG-treated Caco-2 cells (3 μM). TMPRSS2 densitometry is shown, data are mean ± SEM ($n = 3$ biologically independent samples). **E** Immunoblot analysis of Caco-2 cells treated with HFG for 18 h prior to removing the drug, adding fresh media, and then analyzing the protein recovery time course. (All TMPRSS2 densitometry that is shown represents mean ± SEM ($n = 3$ biologically independent samples). Actin is shown as a loading control. $P$-values are shown for comparisons to 0 time point or control, or as indicated by one-way ANOVA with Dunnett's test of multiple comparisons (**B**–**E**).

decreased endogenous TMPRSS2 (Fig. 5E). Moreover, DCAF1 knockdown reduced the accumulation of poly-ubiquitinated TMPRSS2 (Fig. 5F). Lastly, consistent with a direct molecular interaction, we were able to detect DCAF1/TMPRSS2 association by proximity ligation in cells (Fig. 5G).

To determine if halofuginone modulates TMPRSS2 stability through DCAF1, we performed siRNA-mediated knockdown of DCAF1 in the setting of halofuginone treatment. In control knockdown cells, halofuginone reduced TMPRSS2 expression without altering the levels of the cell surface protein E-cadherin (Fig. 6A). In DCAF1-knockdown cells, basal levels of TMPRSS2 increased, and the effect of halofuginone on TMPRSS2 protein levels was abrogated (Fig. 6A). Lysine residues in target proteins are often the site of E3-mediated ubiquitination[28], and TMPRSS2 contains four lysine residues on its intracellular domain, with three lysines closely clustered (K80, K82, and K83). In contrast to the wild-type protein, or to a site-directed mutant in which only two lysines were altered, a site-directed mutant of TMPRSS2 in which either three or four lysines were converted to arginine was resistant to the effects of halofuginone in reducing total TMPRSS2 abundance (Fig. 6B). The 4KR mutant was also expressed on membrane (Fig. 6C) and membrane 4KR-TMPRSS2 was similarly resistant to the effect of halofuginone (Fig. 6D).

Taken together, these results suggest that DCAF1 is a potential E3 ligase subunit that regulates TMPRSS2 stability through ubiquitination, and that the effect of halofuginone on TMPRSS2 levels is in part dependent on the activity of DCAF1.

Of note, halofuginone acts as a glutamyl-prolyl-tRNA synthetase inhibitor, inhibiting translation of a subset of proteins[29]. While halofuginone does not exhibit strict substrate specificity, it has been proposed that halofuginone reduces the rate of translation of "proline-rich" proteins. In that regard, we note that TMPRSS2 sequence contains 7% proline residues, slightly higher than the average of most proteins[30]. Indeed, in addition to altering the stability of TMPRSS2, we observed that halofuginone can inhibit the in vitro and in vivo synthesis of TMPRSS2 in a proline-dependent fashion (Fig. S6a, b). As such, it seems likely that halofuginone can affect both the synthesis and degradation of TMPRSS2.

TMPRSS2 performs a critical Spike protein priming step that facilitates SARS-CoV-2 entry into cells. Since our screens identified agents that reduced TMPRSS2 levels, we asked whether they could prevent pseudotyped SARS-CoV-2 entry. We used a SARS-CoV-2 pseudovirus that faithfully recapitulates SARS-CoV-2 infection, a strategy which has been employed recently by other groups[9,31,32]. In our case, we molecularly tagged the Spike protein with the HiBiT sequence to allow rapid and sensitive detection of

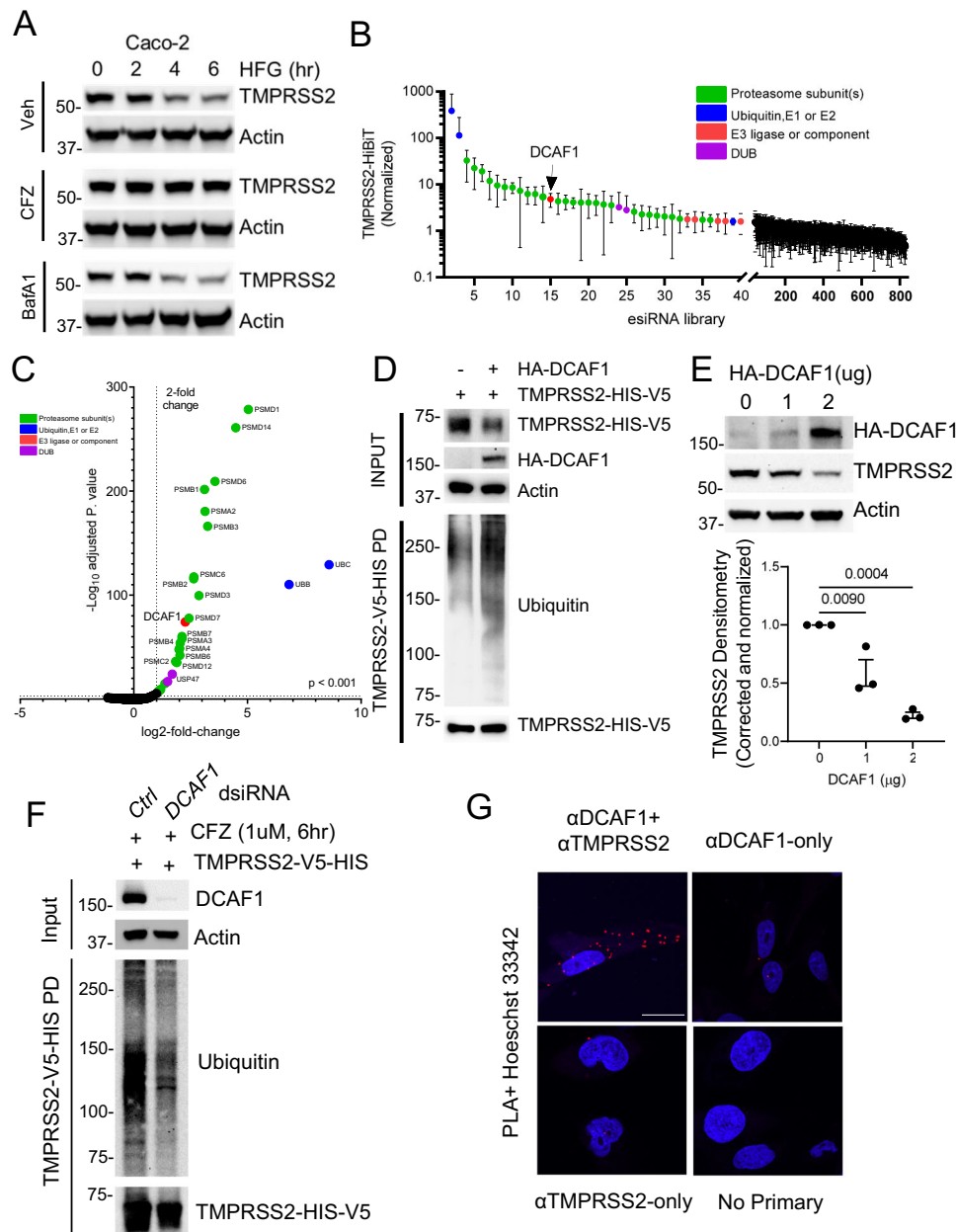

**Fig. 5 TMPRSS2 is degraded through ubiquitin proteasome system, which is required for HFG efficacy. A** Immunoblotting assay of Caco-2 cells treated with HFG (3 µM) in combination with carfilzomib (CFZ, 1 µM) or bafilomycin A1 (BafA1, 1 µM) and probed for TMPRSS2. **B**, **C** TMPRSS2-HiBiT signal was measured following siRNA knockdown of ubiquitination-related machinery. **B** Ubiquitination siRNA library screening results ordered by increase in TMPRSS2-HiBiT signal. The E3 ligase DCAF1 was detected as a top hit. Data are mean ± SD (n = 3 biologically independent samples). **C** Volcano plot of TMPRSS2-HiBiT signal screening with Ubiquitination siRNA library from n = 3 independent screening assays. Statistical significance is plotted against log2-fold change in TMPRSS2-HiBiT signal. Top hits are annotated. **D** DCAF1 co-expression increases TMPRSS2 ubiquitination. **E** Immunoblot analysis of Beas-2B cells with increasing expression of DCAF1, TMPRSS2 densitometry shown represents mean ± SEM (n = 3 biologically independent samples). **F** Immunoblotting of TMPRSS2 ubiquitination following control or DCAF1 siRNA treatment to Beas-2B cells, and TMPRSS2 pull-down. **G** Proximity Ligation Assay between TMPRSS2 and DCAF1 in Beas-2B cells, with nuclear counterstaining by Hoechst33342. Control treatments with antibody removal are also shown. Scale bar = 25 µm. P-values are shown for comparisons to control, or as indicated by one-way ANOVA with Dunnett's test of multiple comparisons (**E**-**F**).

viral infection. The strategy employed is diagrammed in Fig. 7A. We first assessed the ability of our SARS-CoV-2 pseudovirus to infect various cell lines. As previously noted, both Caco-2 and Calu-3 cells appeared to be highly permissive (Fig. 7B), consistent with the known high-level expression of both ACE2 and TMPRSS2 in these cell lines[9,22,33]. We then asked whether homoharringtonine or halofuginone were biologically active in reducing SARS-CoV-2 infection. Remarkably, both agents were

able to affect a 50% reduction in the level of pseudoviral entry in Caco-2 cells at concentration ~300 nM (Fig. 7C, D). Intriguingly, proline augmentation to cell culture media ablated halofuginone's protection against pseudoviral entry (Fig. S6C, D). Similar, albeit slightly less potent effects, were also seen in Calu-3 cells (Fig. S7a, b). We noted that other agents identified in our screen including cilnidipine, dasatinib, and venetoclax were also effective in reducing viral entry (Fig. S7c, d). The pseudovirus also contains a

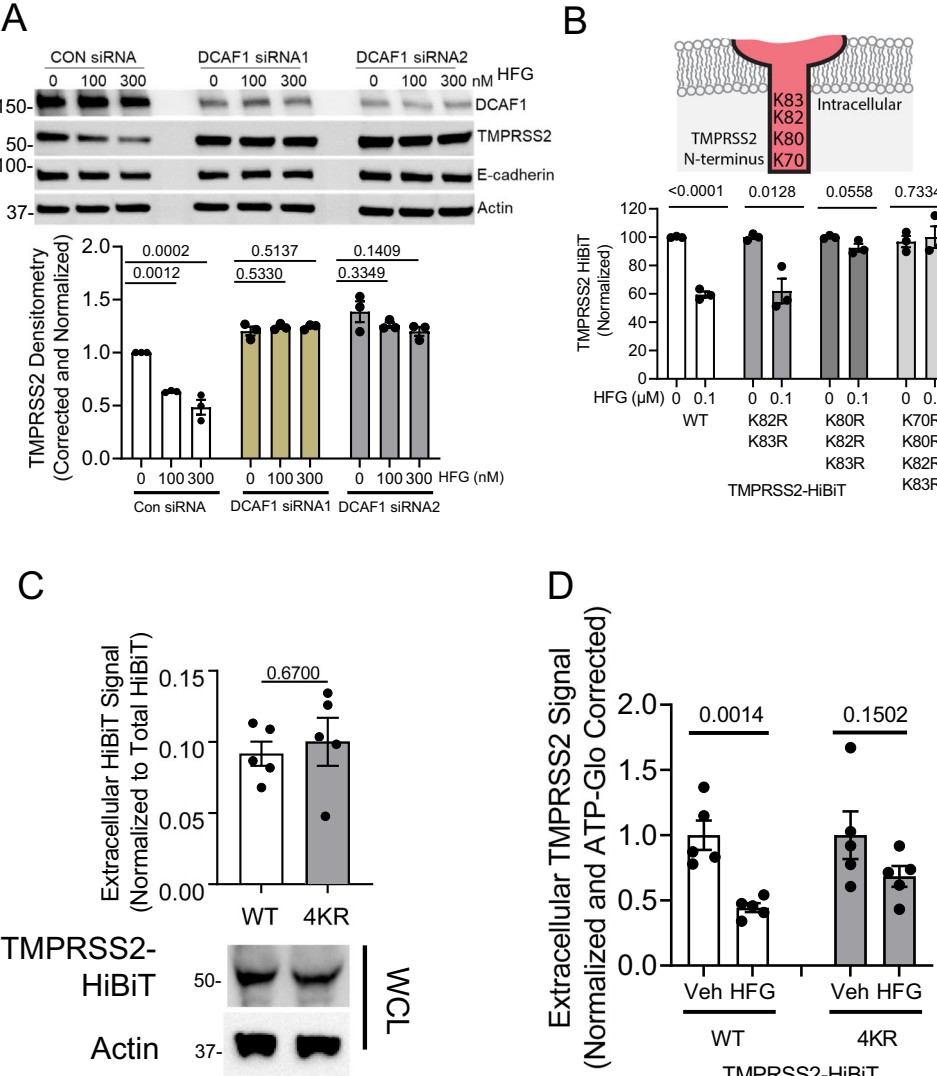

**Fig. 6 Halofuginone effect on TMPRSS2 requires intracellular lysines. A** Immunoblot analysis of Beas-2B cells with control or DCAF1 siRNA treatment followed by halofuginone (HFG) treatment (6 h). TMPRSS2 densitometry represent mean ± SEM ($n = 3$ biologically independent samples). **B** TMPRSS2 intracellular lysines were assayed for their responsiveness to HFG through the HiBiT assay, data represent mean ± SEM ($n = 3$ biologically independent samples). **C** HiBiT-tagged TMPRSS2 WT and 4KR lysine mutant equivalently reached the plasma membrane. Data from extracellular HiBiT assay represent mean ± SEM ($n = 5$ biologically independent samples). **D** HFG treatment reduces WT but not mutant lysine TMPRSS2 protein from reaching cellular membrane. Data from extracellular HiBiT assay represent mean ± SEM ($n = 5$ biologically independent samples). *P*-values are shown for comparisons to control, or as indicated by one-way ANOVA with Dunnett's test of multiple comparisons (**A**) or two-way unpaired *t*-test (**B–D**).

GFP reporter, and as a complementary approach, we observed that both homoharringtonine and halofuginone markedly reduced levels of GFP expression (Fig. 7E), again consistent with their ability to inhibit viral entry. Finally, we tested a separate pseudovirus which encodes for a traditional luciferase-based reporter and showed a similar dose-dependent reduction of the luciferase signal using both homoharringtonine and halofuginone (Fig. S8a, b). In addition, we noted that combined treatment with homoharringtonine and halofuginone was more effective than either agent alone (Fig. S7e).

We next tested whether our identified compounds could prevent pseudotyped SARS-CoV-2 entry in primary human respiratory epithelial cells. Cells were obtained from normal human lungs under a protocol approved by our Institutional Review Board (IRB) and maintained at an air–liquid interface. In culture, these cells retain features of their normal apical-basal polarity and exhibit the expected bronchial epithelial mucociliary phenotype[34,35]. Similar to what we observed in immortalized cell lines, homoharringtonine and halofuginone were also effective in blocking SARS-CoV-2 pseudoviral entry in these primary human bronchial epithelial cells (Fig. 7F, G). Next, we confirmed the role of DCAF1 in regulating the beneficial effects of halofuginone. DCAF1 knockdown abrogated the effect of halofuginone in blocking SARS-CoV-2 pseudoviral entry (Fig. 7H). Similarly, the effects of halofuginone on SARS-CoV-2 pseudoviral entry were abrogated by overexpression of the degradation-resistant 3KR-TMPRRS2 mutant compared to WT TMPRSS2, suggesting the importance of the lysine ubiquitination site in regulating TMPRSS2 stability (Fig. 7I).

Finally, we investigated the effect of homoharringtonine and halofuginone in preventing infection of replication-competent SARS-CoV-2 virus. We utilized the BSL3 facilities at the IIT Research Institute for these assays. Calu-3 cells were pre-treated with either homoharringtonine or halofuginone for 24 h prior to viral entry assay, as previously described[9]. Briefly, treated cells were washed and inoculated with two doses (0.01 MOI or 0.001 MOI) of

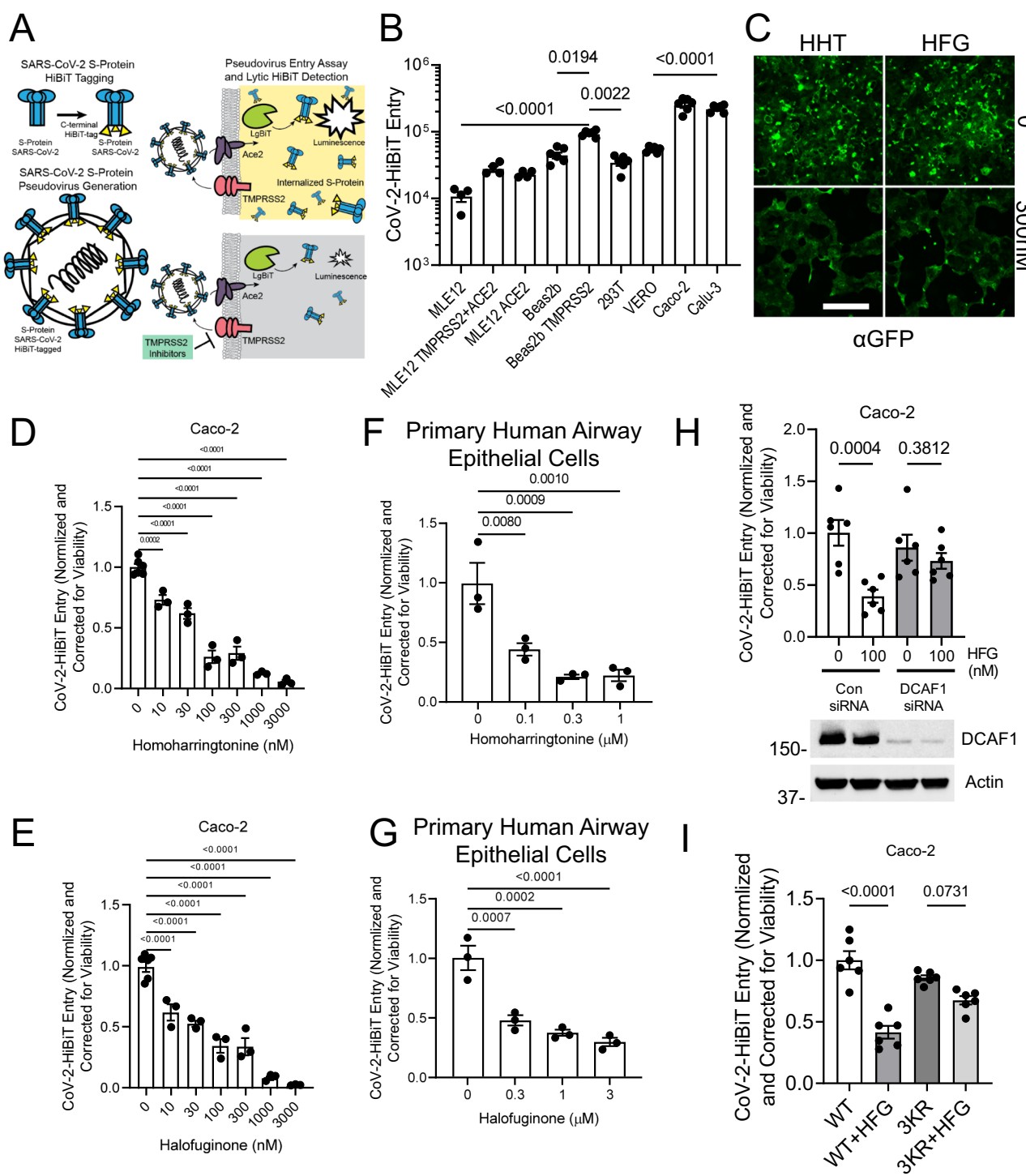

**Fig. 7 Agents that reduce TMPRSS2 expression markedly inhibit SARS-CoV-2 pseudoviral infection. A** Schematic of pseudoviral construction and assay. The S protein of SARS-CoV-2 was C-terminally tagged with HiBiT. **B** Level of viral transduction in various cell lines plotted on a logarithmic scale. Calu-3 and Caco-2 cells had the highest observed rates of infection. Data are mean ± SEM ($n = 4–6$ biologically independent samples). **C** Effects of HHT or HFG on pseudoviral-mediated GFP expression, determined by immunostaining, scale bar = 500 μm. **D, E** Effects of increasing concentrations of homoharringtonine (HHT) (**D**) or halofuginone (HFG) (**E**) on SARS-CoV-2 pseudoviral infection. Data are mean ± SEM ($n = 3–6$ biologically independent samples). **F, G** SARS-CoV-2 pseudoviral infection of primary human bronchial epithelial cells in the presence of increasing concentrations of HHT (**F**) and HFG (**G**). Data are mean ± SEM ($n = 3$ biologically independent samples). **H** SARS-CoV-2 pseudoviral infection of Caco-2 cells transfected with DCAF1 siRNA along with HFG treatment (100 nM). Data are mean ± SEM ($n = 6$ biologically independent samples). **I** SARS-CoV-2 pseudoviral infection of Caco-2 cells transfected with TMPRSS2 WT or lysine mutant prior to HFG treatment (100 nM). All SARS-CoV-2 pseudoviral data is corrected to cell number as determined by CellTiter-Glo. Data are mean ± SEM ($n = 6$ biologically independent samples). *P*-values are shown for comparisons to 0 time point or control, or as indicated by one-way ANOVA with Dunnett's test of multiple comparisons (**F, G**), or a two-way ANOVA with Tukey's test of multiple comparisons (**B, H, I**).

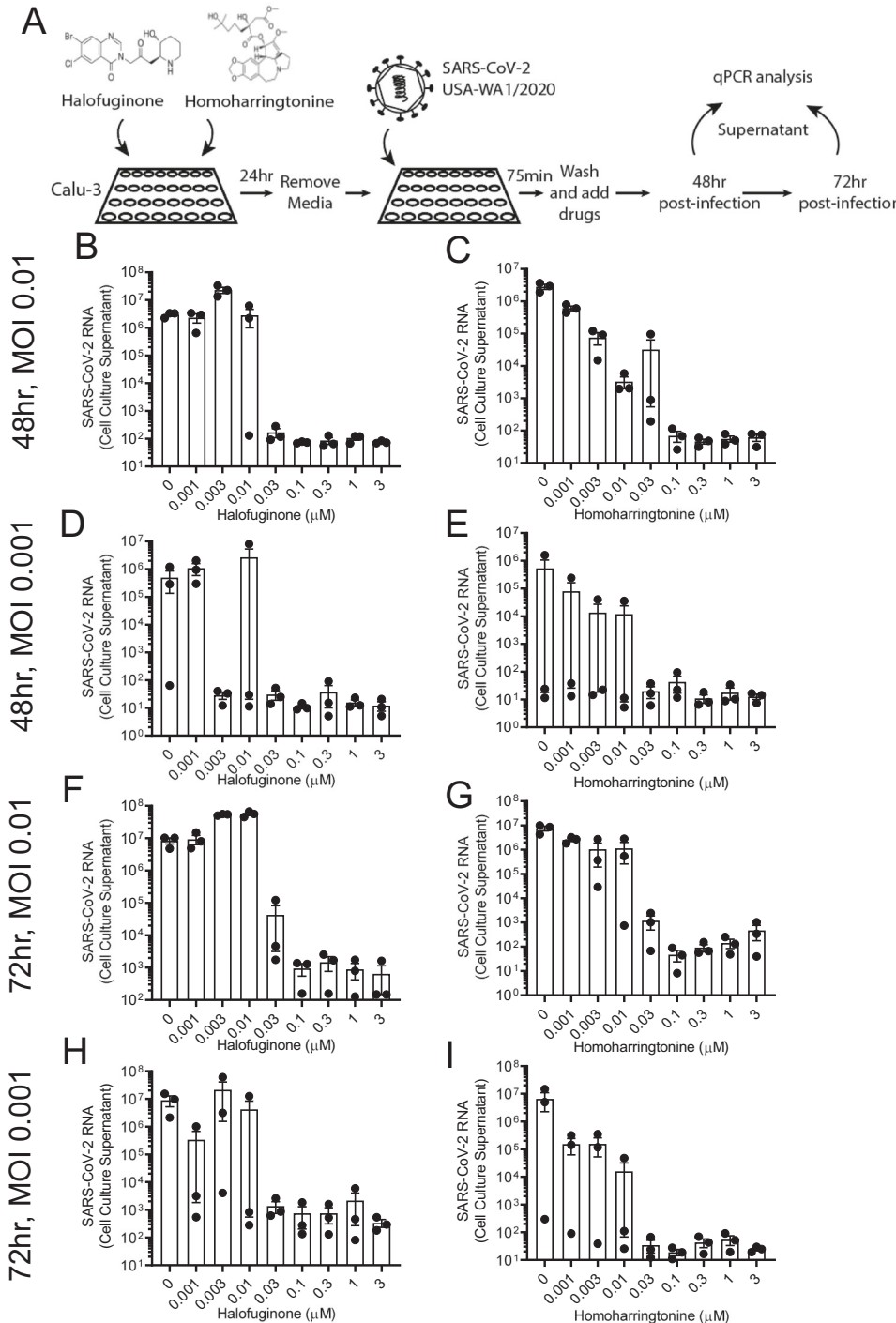

**Fig. 8 Homoharringtonine and Halofuginone show efficacy in blunting SARS CoV-2 infection. A** Schematic of SARS-CoV-2 treatment strategy. Calu-3 cells were pretreated with Halofuginone (HFG) or Homoharringtonine (HHT) for 24 h. Media was exchanged, and cells were exposed to SARS-CoV-2 virus for 75 min, then exchanged for fresh media replenished with HFG or HHT. Supernatant samples were taken at 48 and 72 h for qPCR analysis of viral RNA (**B–I**). **B–I** qPCR analysis of viral RNA in culture supernatant of Calu-3 cells under titration of HFG or HHT at 48 h following an initial infection of SARS-CoV-2 using an MOI of 0.01 (**B–C**), or 0.001 (**D–E**), or at 72 h after infection with an initial MOI of 0.01 (**F–G**) or 0.001 (**H–I**). Data represent mean ± SEM ($n = 3$ biologically independent samples).

SARS-CoV-2 (wildtype-USA-WA1/2020) for 75 min. After the exposure period, cells were washed and fresh media containing homoharringtonine or halofuginone was added. After an additional 48 or 72 h incubation, cell supernatants were collected for qPCR analysis to quantify viral transcript load. We observed that both HHT and HFG had a strong dose-dependent effect in blunting SARS-CoV-2 viral load at both viral MOIs used and at both time points (Fig. 8). At 30 nM, both HHT and HFG were highly effective in reducing SARS-CoV-2 viral infection in these cells.

## Discussion

By executing an unbiased small molecule screen, we have identified a number of compounds that are currently in active clinical

trials or are FDA-approved that can reduce TMPRSS2 expression (Table 1). Using the biological IC50 for inhibiting pseudoviral entry, several of these candidates would appear to be clinically viable. The agents identified in our screen may be effective alone, or in combination with each other (Fig. S7e). In addition, as noted initially, SARS-CoV-2 can enter cells through TMPRSS2-mediated pathways or can employ endo-lysosomal proteases such as cathepsin L to cleave the S protein and gain cell entry[8,36]. Thus, agents that interfere with endolysosomal proteases could be synergistic with agents identified here that inhibit TMPRSS2-mediated pathways (Fig. S12). Our approach would also appear to complement strategies that directly target viral replication such as remdesivir[37]. It should be noted that our two most promising leads, homoharringtonine and halofuginone, have been noted to have anti-viral activity[38–40], although the precise mechanisms for these effects were not previously determined. Our in vitro data would indicate that TMPRSS2 reduction occurs at concentrations that are achievable with the approved dosing in humans (see Table 1).

Homoharringtonine (omacetaxine mepesuccinate) is approved for the treatment of refractory chronic myeloid leukemia in the United States[41]. Homoharringtonine binds to free ribosomes and inhibits protein translation by preventing peptide chain elongation[24]. Thus, it has broad cellular effects. Its effect is most pronounced in the context of malignant cells that display increased rates of protein translation. By depleting very-short lived proteins, malignant cells treated with homoharringtonine undergo apoptotic cell death[42,43]. In addition to its effects on protein translation, other mechanisms of action have been explored. Notably, homoharringtonine has been shown to stimulate degradation of the fusion protein BCL-ABL[44]. Clinically, treatment with homoharringtonine is frequently accompanied by grade III or IV adverse events, including severe myelosuppression[41], thus, limiting its clinical utility at high doses. However, homoharringtonine has also been identified as an agent that can prevent SARS-CoV-2 replication in vitro[45]. In particular, SARS-CoV-2 replication was diminished in homoharringtonine-treated Vero E6 cells with an EC50 estimated at 2.1 μM. Our results show that homoharringtonine acts to prevent viral entry by reducing TMPRSS2 levels, an effect that would not be observed in Vero E6 cells, since this cell line does not express TMPRSS2[46]. Thus, as expected the EC50 we observed in TMPRSS2-expressing Calu-3 cells is in the sub-micromolar range (Fig. 8). Lastly, homoharringtonine has also been identified to have broad anti-viral activity, inhibiting replication of a number of viruses in vitro[38,39], probably via inhibition of viral protein translation.

As noted, homoharringtonine is a chemotherapeutic agent that can trigger myelosuppression[24,25], while halofuginone is generally better tolerated[26,27]. As such, we concentrated our mechanistic studies on the latter, likely more clinically viable agent, demonstrating that TMPRSS2 is targeted for proteasomal-mediated degradation. We have previously described an unbiased siRNA-mediated strategy to rapidly identify which of the more than 600 E3 ubiquitin ligases are relevant for a given target's half-life[47]. Using that platform, we were able to identify DCAF1 as an important regulator of TMPRSS2 stability. Of note, knockdown of DCAF1 raised the basal level of TMPRSS2. More importantly, knockdown of DCAF1 abrogated the ability of halofuginone to trigger a decline in TMPRSS2 levels. Similarly, DCAF1 knockdown reduced the ability of halofuginone to block SARS-CoV2 pseudoviral entry. DCAF1 is a substrate receptor of the Cullin Ring Ligase 4 (CRL4) E3 ligase complex[48]. It functions to connect the CRL4 ubiquitin-transferring machinery to a number of substrate proteins that interact with various motifs within the DCAF1 protein. Published DCAF1 substrates include Merlin[49], RAG1[50], MCM10[51], and NRF2[47]. Importantly, while DCAF1

interacts with these endogenous substrates, it was initially described as a binding partner of the lentiviral protein Vpr[52]. Vpr binds a number of host anti-viral proteins including SAMHD1 and UNG2, and through the interaction with DCAF1, these substrates are ubiquitinated by the CRL4 complex. This mechanism is thought to play a major role in how viruses hijack host protein degradation pathways. The specific role of DCAF1 in the pathophysiology of disease is poorly understood. DCAF1 KO mice are embryonically lethal[53], and thus studies with conditional DCAF1 knockout or transient downregulation have been used to study its function. Most recently, DCAF1 was shown to play a key role regulatory T-cell senescence and ageing[54]. Thus, our results add to the growing body of literature on DCAF1 and implicate it as a factor regulating stability of the host protease TMPRSS2.

Our results suggest that halofuginone catalyzes a reduction of TMPRSS2 protein levels through a DCAF1-dependent pathway. This is further supported by the observation that altering a cluster of critical lysine residues on TMPRSS2's cytoplasmic domain abrogates the effects of halofuginone. Our preliminary data suggests that halofuginone does not inhibit TMPRSS2 catalytic activity (Fig. S9) or directly bind to either TMPRSS2 or DCAF1 (Fig. S10A, B). Similarly, homoharringtonine or halofuginone do not affect DCAF1 gene expression (Fig. S10C). This suggests the drug effects may occur indirectly, perhaps by triggering a post-translational modification of TMPRSS2 that favors an enhanced interaction with DCAF1. One such possible post-translational modification comes from previous observations that DCAF1 binding to a substrate can be triggered my mono-methylation of the substrate, thereby creating a methyl-degron[55]. Whether or not halofuginone can catalyze this or other modifications will require additional study. Of note, halofuginone has also been shown to act as a glutamyl-prolyl-tRNA synthetase inhibitor, inhibiting translation of a subset of proteins[29]. Our data suggests this mechanism may be operative here as well, as proline augmentation to cell culture media ablated halofuginone's protection against pseudoviral entry (Fig. S11A, B). Further, halofuginone treatment was ineffective in preventing viral entry in DCAF1 depleted airway cells (Fig. S11C). As such, it appears possible that halofuginone could affect both the translation and the post-translational stability of TMPRSS2.

While we were focusing on TMPRSS2 in this study, recent work has demonstrated that several other host proteases can perform the critical S protein priming step. Notably, SARS-CoV-2 contains an extra "RRAR" amino acid sequence in the S1/S2 domain, making it susceptible to cleavage by the host protease furin during viral packaging[11–14]. Furin is a highly expressed proprotein convertase that performs vital cellular functions, and the presence of this furin-like cleavage domain in other viruses is associated with increased pathogenicity and neurotropism[56–58]. Future studies are needed to determine whether halofuginone or other promising hits from this study can also reduce furin, or other key host proteases, involved in SARS-CoV-2 entry. Finally, while our results focused on strategies that alter the post-translational stability of TMPRSS2, it should be noted that TMPRSS2 expression is known to be transcriptionally regulated by androgens[59]. It is intriguing to speculate whether this may translate into a higher basal level of TMPRSS2 in men, and in turn, whether this higher level of TMPRSS2 expression can partially explain why men appear to be at significantly higher risk for mortality and complications following COVID-19 infection[60]. Of note, individuals who have inherited a non-coding SNP that only modestly increases the expression of TMPRSS2, appear to be at a significant elevated risk for developing more severe viral infections[61]. As such, we believe that agents identified here, that reduce TMPRSS2 expression, represent a rational approach to modify the clinical course of COVID-19, and potentially future related viral pandemics.

## Methods

**Materials**. High-Capacity cDNA Reverse Transcription Kit (4368814) and SYBR Green PCR Master Mix (4364344) were from Applied Biosystems. BEAS-2B (CRL-9609), Caco-2 (HTB-37), Calu-3 (HTB-55), MLE-12 (CRL-2110), and Vero (CCL-81) cells were all obtained from ATCC. Easy Prep RNA Miniprep Plus Kit (R01-04) was from Bioland Scientific. DC Protein Assay Reagent A/B/S (500-0113/ 0114/ 0115) was from BioRad. GFP (4B10, 2995) and Ubiquitin (4B10, 2955) antibodies were from Cell Signaling Technologies. DMEM/F-12 (11320082), EMEM (670086), Fetal Bovine Serum (26140079), and Opti-MEM I Reduced Serum Medium (31985062) were from Gibco. Amaxa Nucleofector II was from Lonza. Anti-β-actin antibody (MA5-15739), anti-HA-tag antibody (2-2.2.14, 26183), Countess II Automated Cell Counter (AMQAX1000), and Lipofectamine 3000 Transfection Reagents (L3000015) were from Invitrogen. HIV-1 Gag p24 DuoSet ELISA (DY7360-05) was from R&D Systems. SARS-CoV-2 (2019-nCoV) Spike ORF mammalian expression plasmid (Codon Optimized) (VG40589-UT) was from SinoBiological. The following reagent was obtained through the NIH AIDS Reagent Program, Division of AIDS, NIAID, NIH: HIV-1 pNL4-3 ΔEnv Vpr Luciferase Reporter Vector (pNL4-3.Luc.R-E-) from Dr. Nathaniel Landau. pSF-CMV-VSVG (OG592) VSV G Expression Plasmid (SnapFast Pro) (OG592) was from Oxgene. AsiSI (R0630), Eco53kI (R0116S), PmeI (R0560), Quick Ligation Kit (M2200) were from New England Biolabs (NEB). Mouse monoclonal anti–V5 Tag (R960-25) and TMPRSS2 antibody (PA5-83286) were from Thermo Fisher. X-tremeGENE HP DNA Transfection Reagent (6366244001) and X-tremeGENE siRNA Transfection Reagent (4476115001) were from Sigma-Aldrich. Lenti-X GoStix Plus (631280) were from Takara. Venetoclax (HY-15531) and Homoharringtonine (HY-14944) were from MedChemExpress. Halofuginone (S8144) and Cilnidipine (S1293) were from SelleckChem. Bafilomycin A1 (11038), Carfilzomib (17554), Dasatinib (11498), and Verteporfin (17334) were from Cayman Chemical. Antibody against E-cadherin (G-10, sc-8426) was from Santa Cruz Biotechnologies. MG132 (F1100) was from UBPBio. CellTiter-Glo 2.0 Cell Viability Assay (G9243), Nano-Glo HiBiT Lytic Detection System (N3040), Nano-Glo HiBiT Extracellular Detection System (N2421), Nano-Glo HiBiT Blotting System (N2410), HiBiT CMV-neo Flexi Vectors (N2401, N2391) were from Promega. Please see oligonucleotide sequences in Supplementary Table 1. All antibodies were used at 1:2000 dilution unless noted otherwise.

**Cell culture**. Beas-2b and MLE-12 cells from ATCC were cultured in HITES media supplemented with 10% fetal bovine serum (FBS). Caco-2 and HEK293T were from ATCC and cultured in DMEM (Gibco) supplemented with 10% FBS. Calu-3 and Vero cells were from ATCC and cultured in EMEM (ATCC) supplemented with 15% FBS. For the generation of primary bronchial epithelial cells, following attaining informed consent, airway segments and lung tissue were obtained from excess pathological tissue following lung transplantation in accordance with a protocol approved by the University of Pittsburgh Investigational Review Board[35]. The isolation, growth, and maintenance of these cells were as previously described[34,35]. Cells were treated with compound at indicated doses for indicated times, or at the following doses: HFG, 3 μM; Carfilzomib, 1 μM; Bafilomycin A1, 1 μM.

**Cloning**. HiBiT-tagged TMPRSS2 and SARS-2-CoV spike protein plasmid constructs were generated using molecular cloning and the FLEXI system (Promega). Briefly, the open reading frame of the target genes were PCR amplified with restriction sites for AsiSI and PmeI and was cloned into pFC37K-HiBiT plasmid. Point mutants (Lysine→Arginine) were generated through QuikChange II XL Site-Directed Mutagenesis Kit (Aglient). All plasmid constructs were verified by DNA sequencing (Genewiz).

**Transfection**. Plasmid transfections were conducted using nucleofection in Beas-2b and MLE-12 cells using Nucleofector II (Amaxa). X-tremeGENE HP DNA transfection reagent or Lipofectamine 3000 transfection reagent was used for plasmid transfections of HEK293T cells.

**FDA-approved compound library screening**. Human bronchial epithelial Beas-2b cells were selected for initial screens given their ease of transfection and ability to model several airway diseases. Beas-2b cells transiently expressing TMPRSS2-HiBiT (C-terminal) were seeded to a final density of $1 \times 10^4$ cells per well. The FDA-approved compound library (100nL per drug) was stamped to 384-well tissue culture plates using CyBio Well vario (Analytik Jena). Compounds were plated to the final concentrations of 10 μM. After 18 h of treatment, culture media was removed and cells were processed for Nanoluciferase activity using Extracellular HiBiT detection system (Promega), according to manufacturer's protocol. After reading the extracellular HiBiT signals, TritonX-100 solution was added into each well (final concentration 0.05%) for cell lysis, and HiBiT signals were acquired again for all plates. Signals were collected and quantified using a Cytation 5 plate reader from Biotek. For secondary screening to determine drug IC50, specific compounds were cherry-picked using a TTP Mosquito X1 followed by serial dilutions of compounds were prepared using a Bravo automated liquid handling platform (Agilent). Cell seeding and extracellular and lytic HiBiT signal acquisition were performed according to the same protocol described above. IC50 and CC50

values for tested compounds were calculated using non-linear regression in GraphPad vs9.

**High-throughput liquid handling**. A Thermo Scientific custom HTS platform and Agilent Bravo automated liquid-handling platform was used to transfer contents of a FDA-approved compound library into assay plates. Biotek EL406 washer dispenser was used to distribute reagents or cell solutions into assay plates. For multiple plates operation, plate and liquid handling sequence and intervals were controlled through the Agilent VWORKs and Thermo Momentum software.

**Cell viability assessment**. Cell viability was tested using CellTiter-Glo 2.0 Cell Viability Assay (Promega). 20 μl reagent was dispensed directly into each well of the 384-well tissue culture plates prior to luminescence signal acquisition by Cytation 5 plate reader.

**Compound washout assays**. Cells were pre-treated with indicated compound for 18-h prior to 1 round of washing and incubation with fresh culture media for the indicated time periods. After the indicated times, cells were collected and processed for TMPRSS2 immunoblotting.

**RT-qPCR**. Total RNA was extracted using RNA Extraction Miniprep Kit from Bioland Scientific, following the manufacturer's protocol. cDNA was prepared using High-Capacity RNA-to-cDNA Kit from Applied Biosystems. SYBR Green Real-Time PCR Master Mixes from Applied Biosystems were used in qPCR, detecting the expression level TMPRSS2, HPN, ST14, and CORIN.

**Ubiquitination assay**. TMPRSS2-V5-HIS in pcDNA3.1D was co-expressed with HA-DCAF1 in pcDNA3 in Beas-2b cells for 18 h, prior to lysis and precipitation with Dynabead HIS-resin (Thermo). Precipitate was eluted in 1xLaemmli Protein Sample Buffer at 95 °C for 10 min, and resolved through SDS-PAGE immunoblotting.

**Ubiquitination siRNA screen**. TMPRSS2-HiBiT cells were screened with an siRNA library targeting Ubiquitination-related machinery[47]. Briefly, 25 ng of siRNA was mixed with Lullaby transfection reagent (OzBiosciences) and diluted in Opti-MEM media. The transfection mixture incubated at room temperature for 20 min, and was added to 20 μL of HITES + 10%FBS media with 2000 cells. Following 72 h knockdown, Lytic HiBiT luciferase assays were performed using manufacturer's protocol. For conformational specific gene silencing, small interfering RNAs were selected and purchased from IDT, and transfected in cells using Lullaby siRNA transfection reagent, with Negative Control DsiRNA transfected as control. Subsequent analysis was performed after 72 h of knockdown.

**In vitro transcription and translation proline treatment assay**. TMPRSS2 constructs were synthesized in vitro using TnT Coupled Reticulocyte Lysate System (Promega) following manufacturer's protocol. HFG plus proline assays were conducted using in vitro transcription and translation kit with free amino acids diluted 5-fold and proline supplemented at indicated concentrations[29]. Following synthesis, products were analyzed via immunoblotting.

**Cell-based proline treatment assays**. Indicated cell line were treated with exogenous proline and compound treatment prior to collection for pseudoviral entry assay or TMPRSS2-HiBiT signal determination.

**Immunoblotting**. Cells were lysed in RIPA buffer supplemented with EDTA-free protease inhibitor tablet on ice. Cell lysates were sonicated at 20% amplification for 12 s and centrifuged at 12,000 g for 10 min at 4 °C. Supernatants were collected and normalized for the total protein concentrations, mixed with 6X protein sample buffer, and incubated at 42 °C for 10 min. Sample lysate was resolved using 4–20% acrylamide PROTEAN® TGX™ precast gels from BioRad and electrophoresed in TGS buffer. The proteins were then electro-transferred to nitrocellulose membranes. Blots were incubated in 15 ml of blocking buffer for 1 h at room temperature, before incubation in 10 ml of the primary antibody solution (1:2000 dilution) overnight at 4 °C. Afterwards, three 10-min washing were performed in 15 ml TBST. Blots were then incubated with 10 ml of the secondary antibody solution for 1 h at room temperature. After three 10-min washing in 15 ml TBST, blots were then developed using West Femto Maximum Sensitivity Substrate from Thermo Scientific, and imaged using ChemiDoc Imaging System from Bio-Rad.

**HiBiT blotting**. Samples from cells transfected with HiBiT-tagged proteins were prepared following the same protocol as immunoblotting. Proteins were transferred to a nitrocellulose membrane, followed by gentle rocking in TBST to rinse away transfer buffer. Nano-Glo HiBiT blotting system (Promega) was used for development, following manufacturer's protocol. Briefly, the blot was incubated in 5 ml 1× Nano-Glo blotting buffer supplemented with 25 μl LgBiT protein overnight at 4 °C. The next day, 10 μl Nano-Glo luciferase assay substrate was directly added

into the solution and mixed well immediately. After incubation for 5 min at room temperature in dark, the blot was imaged by ChemiDoc Imaging System (Bio-Rad), using chemiluminescence mode.

**Pseudovirus entry assays.** Pseudovirus with SARS-CoV-2 spike protein with a C-terminal HiBiT tag was generated by co-transfection of 293 T cells with psPAX2 (Addgene, MA), pLenti-c-mGFP (Origene, MD), and PFC37K-HIBIT-SARS-CoV-2-S (Backbone from Promega, WI) by using lipofectamine 3000 (Invitrogen, CA). Briefly, 293 T cells were seeded one day before in 8 ml DMEM complete media without antibiotics in a 10 cm tissue culture dish. The following morning, cells were transfected with 8 µg psPAX2, 8 µg pLenti-c-mGFP or pLenti-c-FLUC and 4 µg PFC37K-HIBIT-SARS-CoV-2-S with lipofectamine 3000 according to the manufacturer's protocol. Six hours later, media was changed with fresh full media without antibiotics. The supernatants were harvested 48 h post-transfection, and centrifuged at $500 \times g$ for 10 min to remove cell debris. Virus titer was checked by Lenti-X GoStix Plus (Takara,CA). HiBiT expression was checked by Nano-Glo® HiBiT Lytic Detection System (Promega, WI). The HIV-1 Gag p24 content in the produced SARS-CoV-2-HiBiT pseudovirus was quantified by ELISA (R&D systems), following the manufacturer's protocol. To detect infectivity, various cell types with control or compound treatment were incubated for 1 h with pseudovirus, after extensive washes with PBS, cells were lysed with Nano-Glo HiBiT lytic reagent (Promega) for 20 min. Luminescence signals were then acquired by ClarioSTAR microplate reader (BMG Labtech, Cary, NC).

**Immunocytochemistry.** Caco-2 cells were seeded in 384-well glass-bottom plates (Cellvis, 5000 cells/well) and treated with halofuginone or homoharringtonine at the indicated concentrations for 18 h. Cells were then treated with a pseudovirus encoding the SARS-CoV-2 Spike protein and mGFP. Forty-eight hours after infection, cells were fixed (4% paraformaldehyde), permeabilized (0.5% Triton X-100), and stained for GFP (Cell Signaling Technology), followed with a goat anti-rabbit Alexa Fluor 568 secondary antibody. The fluorescent signal was imaged using Image Express (Molecular Devices) to measure viral entry.

**Duolink proximity ligation assay.** Proximity Ligation Assay was conducted using Duolink technology, according to manufacturer's protocol (Millipore-Sigma)[62]. Briefly, Beas-2B cells were seeded to 96-well glass bottom plate (Cellvis) and fixed with 4% paraformaldehyde for 1 h. Cells were permeabilized with 0.5% Triton-X-100 for 0.5 h, and blocked with Duolink Blocking solution at 37 °C for 2 h. Anti-TMPRSS2 (Thermo, PA5-14264) and anti-DCAF1 (Santa Cruz Biotechnology, (C-8, sc-376850) antibodies were incubated with cells overnight. PLA probe incubation (secondary antibody), Ligation, and Amplification were conducted according to protocol. Cells were counterstained with Hoescht 33342 (Invitrogen). Representative confocal images were taken using Leica SP8 confocal.

**SARS-CoV-2 infection assay.** Human lung cancer (Calu-3) cells were maintained in Eagles's Minimum Essential Medium with 10% fetal calf serum. Cells were cultured in 96-well plate to reach 80–90% confluency. Test articles were tested against wildtype USA-WA1/2020 (SARS-CoV-2) in triplicate. Compounds were serially diluted and cells pretreated for 24 h. After pretreatment, the test articles were removed and then incubated with a standardized virus concentration at 37 °C ± 2 °C in 5.0% ± 1% $CO_2$ for 75 ± 15 min. Following the 75 ± 15 min incubation, virus inoculum were removed, cells washed and appropriate wells overlaid with test or control articles in 0.2 mL EMEM2 (EMEM with 2% FBS) and incubated in a humidified chamber at 37 °C ± 2 °C in 5 ± 2% $CO_2$. At 48 ± 6 and 72 ± 6 h post inoculation, from each plate, 120 µl of the supernatant from each well was collected for subsequent analysis by RT-qPCR. The concentration of virus in the cell culture supernatants was determined by qRT-PCR. Briefly, samples kept at ≤ −65 °C were thawed and centrifuged to remove cellular debris. RNA was extracted using the Quick-RNA Viral Kit (Zymo Research) according to the manufacturer's protocol. qRT-PCR was performed using the following RTPCR cycling conditions: 50 °C for 15 min (RT), then 95 °C for 2 min (denature), then 40 cycles of 10 s at 95 °C, 45 s at 62 °C. Virus titer by RT-qPCR was performed according to IITRI SOPs. Primers used for SARS-CoV-2 detection: 2019-nCoV_N1-F 5′-GACCCCA AAATCAGCGAAAT-3′ 2019-nCoV_N1-R 5′-TCTGGTTACTGCCAGTTGAA TCTG -3′ Probe: 2019-nCoV_N1-P: 5′-FAM-ACCCCGCATTACGTTTGGTGG ACC-BHQ1-3′.

**TMPRSS2 activity assay.** Test compounds were pre-incubated with recombinant TMPRSS2 (AA 106-492, LSBio) in assay buffer (TBS, 0.05% Tween-20) for 15 min before addition of Boc-Gln-Ala-Arg-MCA (Peptides International) substrate to a final concentration of 10 µM. Final concentration of TMPRSS2 was 1.35 µg/mL. Fluorescence (380-15 nm excitation/470-20 nm emission) of the samples was monitored over 4:45 h and the slope of the curves was calculated by linear regression analysis and used as output for TMPRSS2 activity.

**Cellular thermal shift assay.** B2B cells were transfected with TMPRSS2-HiBiT plasmid overnight, prior to treatment with vehicle or HFG (3 µM for 1 h). Cells were collected and resuspended in 10 ml of PBS supplemented with EDTA-free protease inhibitor tablet. Cell solutions were aliquoted to 10 PCR microtubes evenly. Using the PCR thermocycler generating temperature gradient, each aliquot was incubated at a certain temperature between 40 and 58 °C with 2 °C interval for 3 min, then at room temperature for 3 min. Samples were immediately snap frozen in liquid nitrogen and 2 cycles of freeze–thaw were followed. After vortex briefly, samples were transferred to 1.7 ml microcentrifuge tubes for centrifuging at 20,000 g, 4 °C for 20 min. Supernatants were carefully acquired and used for subsequent immunoblotting analysis. The whole procedure followed the protocol described in previous literature[63].

**Statistics.** Statistical comparisons were performed in GraphPad Prism. Unpaired two-tailed Student's $t$ test was used to compare two groups. Comparisons of more than two groups were tested with one-way ANOVA with Tukey's post-hoc test of multiple comparisons.

**Reporting summary.** Further information on research design is available in the Nature Research Reporting Summary linked to this article.

## Data availability
The datasets generated during and/or analyzed during the current study are available from the corresponding author on reasonable request. Source data are provided with this paper.

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

## Acknowledgements

This work was supported by NIH grants to B.B.C. (5R35HL139860 and 5R01HL133184), Y.L. (5R01HL142777), J.W.E. (1K08HL144820), T.B.L. (T32 HL110849), J.F.M. (R01 133184), and T.F. (1R01 HL142663, 1R01HL142589, and P30 AG024827), the University of Pittsburgh Aging Institute seed fund to B.B.C., T.F., and Y.L. and a grant from Jewish Healthcare Foundation (TF). We thank Cystic Fibrosis Research Development Program at the University of Pittsburgh, School of Medicine, Pittsburgh, PA for providing HBE cells.

## Author contributions

B.B.C., T.F., Y.L. designed and directed the study. Y.C., T.B.L., J.W.E., B.B.C., T.F. analyzed the data, prepared the figures, and wrote the manuscript. Y.C., T.B.L., J.W.E., M.B.L., I.A., B.L., J.R.K., L.A.S., K.C.L., F.T., J.L. performed all experiments. M.M.M. and J.F.M. provided help with the human bronchial epithelial cells. D.P.C. assisted with high-throughput screening. B.B.C., Y.L., T.B.L., J.W.E., and T.F. provided funding for the studies.

## Competing interests

The authors declare no competing interests.
