## [Peer Review File · Nature Communications]

Reviewers' Comments:

Reviewer #2:

Remarks to the Author:

Drug screening against the pandemic SARS-CoV-2 is one of the most important research to open the new lifestyle with COVID19. Therefore, the clinically active and FDA-approved drugs are big candidates for quick supply to clinical trials for the huge numbers of COVID19 patients. SARS-CoV-2 virus is thought to use ACE2 and TMPRSS2 proteins in the early infection step. In this manuscript, the authors have attempted to identify inhibitor(s) to block an entry of SARS-CoV-2 virus into the cells from approximately 1,200 clinically active and FDA-approved drugs as a target of TMPRSS2. Authors made a cell-based assay system using HiBiT-fused TMPRSS2 for the screening, and subsequently found that the two compounds, homoharringtonine (HHT) and halofuginone (HFG), reduced the level of TMPRSS2 protein in the several cell lines. These compounds also inhibited an infection of SARS-CoV-2 pseudovirus. Furthermore, authors challenged to be clear an inhibitory mechanism of TMPRSS2 reduction by HFG using protease inhibitors and siRNA library focused on ubiquitination. From these experiments, authors found that HFG provided a proteasomal degradation of TMPRSS2 protein and DCAF1 induced the degradation.

Addressing point in this manuscript is quite good. Funding of DCAF1-dependent TMPRSS2 degradation shows high potential for understating the regulation of SARS-CoV-2 infection. However, a biological relationship among TMPRSS2, DCAF1, and HFG is quite unclear. This manuscript therefore seems an incomplete version. For readers of Nature communications, authors should provide evidence for an inhibitory mechanism linking the three components.

Major issues

1. In Figure 5A, authors should use NEDD inhibitor, such as MLN4924, to confirm that TMPRSS2 is degraded by the cullin complex.

2. In Figure 5B and C, authors found many candidate clones that decrease signal of TMPRSS2-HiBiT. However, authors selected DCAF1 without adequate experimental confirmation. Authors experimentally should indicate why other candidate clones were not selected.

3. In Figure 5, authors should show whether siDCAF1 decreases ubiquitination on TMPRSS2.

4. In Figure 5G, signal reduction of TMPRSS2-HiBiT by HFG was rescued by TMPRSS2-K/R mutations. Authors should indicate whether expression of wild type, TMPRSS2-K82R/K83R, or -K80R/K82R/K83R effects on infection of SARS-CoV-2 pseudovirus.

5. Do TMPRSS2-K/R mutations increase signal of extracellular TMPRSS2-HiBiT?

6. Does HFG effect on transcription level of DCAF1?

7. Authors found DCAF1 by siRNA screening using Beas-2b cells. In Figure 6B, many cell lines have accepted infection of SARS-CoV-2 pseudovirus. Authors should show whether siDCAF1 increases the infection in these cell lines.

8. In Figure S6, proline treatment recovered the effect of HFG in vitro translation of TMPRSS2 and TMPRSS2-HiBiT signal. Can the proline treatment cancel the HFG effect for infection of SARS-CoV-2 pseudovirus in many types of cell lines?

Minor issues

1. Chemical structures of HHT and HFG help readers to understand the study.

2. Please explain why authors use Beas-2b cells for the screening?

3. Please explain inhibition mechanism of HHT in discussion.
4. Please discuss biological roles of DCAF1 in other infection diseases.

Reviewer #3:

Remarks to the Author:

Yanwen Chen & al. developed an original assay to identify small molecules that inhibit the expression of TMPRSS2, a cellular protease essential for SARS-CoV-2 entry into target cells. In total, 2700 compounds were screened and 24 candidate molecules were selected for validation. The best candidates, homoharringtonine (HHT) and halofuginone (HFG), were further investigated for their ability to suppress TMPRSS2 expression and impair viral entry. Interestingly, halofuginone has been shown to modulate TMPRSS2 levels through DCAF-1 ubiquitination and proteasomal-mediated degradation, providing insights on the mode of action of this molecule. This study is well performed, original, and conclusions are mostly supported by the experimental data presented.

Comments:

- Fig. S1: Please provide the signal/background ratio for the assay. Define CXH (cycloheximide?) and provide CHX concentration used in the pilot experiment
- Where does the FDA-approved library come from? Please provide the information.
- Normalized HiBit signals are presented in Fig. 2. Please explain the normalization procedure. How many control wells were present in each screening plate?
- Information is missing on the toxicity of tested molecules, thus impeding the interpretation of Fig. 2. Was the NanoLuc signal inhibited because TMPRSS2 is selectively suppressed or because cells were overly stressed and dying. For example, Halofuginone is highly toxic at 10 microM as shown in Fig. 3. Although toxicity assays were performed on the selected molecules in a second time, which is fine, this is a limitation of the initial assay which should be clearly stated in the text.
- It is said that in total 24 compounds were selected from the screen and retested (see the main text). Dose-response curves and validation experiments are presented for only 16 compounds (6 in Fig. 2 and 10 in Fig. S2). Please explain. Were the other compounds toxic or failed to retest?
- Fig. 3 and S2. Inhibition curves with IC50s are presented. Were the observed inhibitions statistically significant? P-values must be provided.
- Fig S2. The scale of the graph showing Costunolide effect on Lytic HiBit signal needs to be corrected and should range from 0-150%.
- "These agents were also directly assessed by Western blot analysis for TMPRSS2-HiBiT protein expression (fig. S3)". This sentence should be rephrased because only homoharringtonine, halofuginone and venetoclax were analyzed by WB.
- Fig. 4. Homoharringtonine or halofuginone reduce TMPRSS2 but not actin or LacZ-V5 expression in lung cells. Other membrane proteins such as E-cadherin (as displayed in Fig. 5f for HFN) should be presented to validate the relative specificity of tested compounds for TMPRSS2
- In line with the previous comment, ACE2 expression in the presence of HHT or HFN should be analyzed. This is essential to validate conclusions from Fig.6.
- Fig. 5D. A plasmid encoding TMPRSS2-HIS-V5 is transfected, but TMPRSS2-V5 expression is indicated on the WB. This needs to be corrected.
- Fig. S5. Please provide cellular viability for Calu-3 cells when treated with HHT or HFG.
- Fig. 6A. The binding of viral particles tagged with the HiBiT peptide is expected to generate background signal when using the lytic reagent. This point needs to be addressed. Explain how viral binding to cellular membrane is distinguished from actual viral entry.
- Fig. 6c and d. Please indicate that presented experiments were conducted in Caco-2 cells.
- In their conclusion, authors should clearly indicate that in addition to TMPRSS2, HHT and HFN potentially impair the expression of other membrane factors involved in viral entry. This cannot be excluded at this point.

Reviewer #4:

Remarks to the Author:

In this manuscript, Chen et al. use a repositioning screen of a library of FDA-approved or clinically tested drugs to identify compounds that downregulate TMPRSS2 expression, and thereby limit SARS-CoV-2 entry. The focus of this study is of great interest in the current pandemic as effective antivirals against SARS-CoV-2 remain an unmet clinical need. The repurposing of clinically tested drugs could allow a rapid transition to an in vivo evaluation, and the focus on TMPRSS2 expression uses a different approach than similar screens.

The study identifies two lead molecules and proposes a new mechanism of action of halofuginone, the most promising compound, based on DCAF-1-mediated proteasomal degradation.

To my knowledge, antiviral activity of halofuginone against SARS-CoV-2 has not previously been described. Moreover, the insights into a potential mechanism of action for the antiviral activity of this compound are novel. The study is generally well controlled and statistical analyses are applied appropriately. However, I do think that the description of individual experiments could be improved by providing additional details. In cases where the authors refer to previous publications, providing a brief summary of the experimental approach would be helpful for the reader.

While I support the publication of this study in general, I have the following suggestions for the authors to strengthen the manuscript:

A current weakness of the paper is the lack of detailed information on both the setup/analysis and the results of the compound screen.

1) The results of the screen should be made available in their entirety. Drug repurposing datasets, such as the one generated in this study, are of great value for drug discovery efforts and secondary analyses that are particularly relevant in light of the current pandemic. Unless there are compelling reasons precluding publication of the complete dataset, it should be made available as supplementary information or through a public data repository.

2) Additional details describing the setup and data analysis of the screen should be provided. What positive and negative controls were used (Fig S1 shows "Veh" and "CHX", but no explanation is provided)? How was the data normalized? Were replicates included in the screen, and if so, what was the correlation?

3) Showing the controls (positive and negative) in Figures 2A-C would help the reader to assess the impact of the identified compounds.

4) The manuscript currently does not provide a clear narrative how 24 compounds were selected out of the ~100 hits, and which ones could be validated. A supplementary table/file showing the screening results should be provided for all ~100 identified compounds and indicate which compounds were selected for follow up; it is currently unclear which 24 compounds were chosen and why.

5) Only 16 of 24 selected compounds are shown in dose response (Fig 3, S2) and 3 by western blot (Fig S3). This data needs to be shown for all 24 compounds and indicate which compounds are considered active. This will also allow an assessment of the screen reconfirmation rate.

The dose response curves in Fig 3 and S2 do not seem to fit the data in all cases. Particularly for Dasatinib and Docetaxel the respective fit is poor and unlikely to provide a correct IC50. In other cases, including Homoharringtonine and Halofuginone, the curve fit suggests a stronger maximal effect than the measured dose response data supports, which may affect the calculated IC50.

6) How were the IC50 values calculated, and could a model with additional parameters (e.g. a four-parameter logistic equation) provide a better fit?

7) The text indicates that the drugs' toxicity (CC50) and SI were determined. These values should be provided whenever a CC50 could be determined.

The results section describes that measured IC50 values were compared to published PK properties, which I assume is referring to Table 1.

- 8) If so, I recommend referring to Table 1 in the results section and clearly state from which experiment the listed IC50 values are derived.
- 9) I was unable to verify the PK data provided in Table 1 since the references appear to be incorrect or mixed up; please update the references.

An important caveat of the manuscript is the absence of infection assays with live SARS-CoV-2 that confirm the activity of the compounds in the context of viral infection and replication. It should be noted that the activity of homoharringtonine against SARS-CoV-2 has been investigated in a recent peer-reviewed study. Choy et al. (<https://doi.org/10.1016/j.antiviral.2020.104786>) report an EC50 for the inhibition of SARS-CoV-2 by homoharringtonine in Vero E6 cells of 2.1 μ M, supporting this compound's activity. Nevertheless, and despite the differences in experimental systems, the data raises the question to what extent the results of the pseudovirus entry assay conducted in this study (\sim 30nM) are predictive of the activity against SARS-CoV-2 viral infection.

- 10) Since an infection with live SARS-CoV-2 could not be performed, it would be helpful to discuss these potential caveats. This is especially relevant given the narrow predicted therapeutic window for the lead compounds when comparing the IC50 data to available PK properties.

The authors propose a mechanism of action for halofuginone based on DCAF-1-mediated proteasomal degradation of TMPRSS2. In addition, they show evidence for a second, previously described MOA based on the inhibition of prolyl-tRNA synthetase. However, the contribution of these mechanisms to the downregulation of TMPRSS2 by halofuginone remains unclear and further insight into their biological relevance would be of great interest.

- 11) The data shown in Fig S6B seems to indicate that the addition of proline completely abolishes the effect of HFG on TMPRSS2 expression. If rescuing prolyl-tRNA synthetase inhibition eliminates the effect of HFG on TMPRSS2 levels, what is the relevance of HFG-mediated proteasomal degradation of TMPRSS2? Could the inhibition of pseudoviral entry be reversed by the addition of proline?

The data in Figure S10 appears too preliminary for publication and should be revised or removed.

- 12) The experiment needs to be conducted in replicates, including a statistical evaluation.

Additional minor points:

- 13) What is the rationale for the selection of compounds highlighted in Fig 2C, since they do not seem to match those listed in Fig 2A/B?
- 14) In Fig 3, please define what the dotted lines along the modeled dose response curve represent.
- 15) Fig 4A: What compound concentrations were used?
- 16) Fig 4B: How was the screen normalized and what controls were used? A brief summary of the experimental setup in addition to referencing the original publication would be helpful.
- 17) The label for Fig 4C is missing.
- 18) Fig 4F: What timepoint does the data represent?
- 19) Fig 6H: A (representative) western blot showing the knockdown level of DCAF1 should be included.
- 20) Statistics need to be added to Fig S6B, as well as a brief description of the experimental setup.

REVIEWER COMMENTS

Reviewer #1 (Remarks to the Author):

Drug screening against the pandemic SARS-CoV-2 is one of the most important research to open the new lifestyle with COVID19. Therefore, the clinically active and FDA-approved drugs are big candidates for quick supply to clinical trials for the huge numbers of COVID19 patients. SARS-CoV-2 virus is thought to use ACE2 and TMPRSS2 proteins in the early infection step. In this manuscript, the authors have attempted to identify inhibitor(s) to block an entry of SARS-CoV-2 virus into the cells from approximately 1,200 clinically active and FDA-approved drugs as a target of TMPRSS2. Authors made a cell-based assay system using HiBiT-fused TMPRSS2 for the screening, and subsequently found that the two compounds, homoharringtonine (HHT) and halofuginone (HFG), reduced the level of TMPRSS2 protein in the several cell lines. These compounds also inhibited an infection of SARS-CoV-2 pseudovirus. Furthermore, authors challenged to be clear an inhibitory mechanism of TMPRSS2 reduction by HFG using protease inhibitors and siRNA library focused on ubiquitination. From these experiments, authors found that HFG provided a proteasomal degradation of TMPRSS2 protein and DCAF1 induced the degradation.

Addressing point in this manuscript is quite good. Funding of DCAF1-dependent TMPRSS2 degradation shows high potential for understating the regulation of SARS-CoV-2 infection. However, a biological relationship among TMPRSS2, DCAF1, and HFG is quite unclear. This manuscript therefore seems an incomplete version. For readers of Nature communications, authors should provide evidence for an inhibitory mechanism linking the three components.

Major issues

1. In Figure 5A, authors should use NEDD inhibitor, such as MLN4924, to confirm that TMPRSS2 is degraded by the cullin complex.

Response: We have included additional data in the manuscript (Figure S5C) examining whether MLN4924 similarly reduces TMPRSS2 levels. As this is an inhibitor of the NEDD8 E1 activating enzyme, and we identify the Cullin-Ring ligase subunit DCAF1 as a potential controller of TMPRSS2 protein stability, we hypothesize that MLN4924 treatment would preserve TMPRSS2 protein by inactivating DCAF1 activity. Indeed, we observe that co-treatment of MLN4924 with HFG leads to a preservation of TMPRSS2 protein level. The data is shown below:

2. In Figure 5B and C, authors found many candidate clones that decrease signal of TMPRSS2-HiBiT. However, authors selected DCAF1 without adequate experimental confirmation. Authors experimentally should indicate why other candidate clones were not selected.

Response: We conducted three replicates of the ubiquitination siRNA library experiment with TMPRSS2-HiBiT cells. From these, the only significantly associated ubiquitin E3 ligase was DCAF1. All other top hits were ubiquitin, proteasomal subunits, or de-ubiquitinase enzymes; the other 'top' hits for E3 ligases were statistically insignificant. We have adjusted our volcano plot of the ubiquitination siRNA experiment to illustrate the different types of hit proteins, as well as our threshold for consideration (2-fold change in TMPRSS2 signal, and significance of $p < 0.001$). These are marked as lines on the plot. From this, we proceeded with DCAF1 as our candidate E3 ligase.

3. In Figure 5, authors should show whether siDCAF1 decreases ubiquitination on TMPRSS2.

Response: We have performed the experiment as requested by the reviewer and included it as new data in Figure 5F. We observed that DCAF1 KD reduces TMPRSS2 ubiquitination:

4. In Figure 5G, signal reduction of TMPRSS2-HiBiT by HFG was rescued by TMPRSS2-K/R mutations. Authors should indicate whether expression of wild type, TMPRSS2-K82R/K83R, or -K80R/K82R/K83R effects on infection of SARS-CoV-2 pseudovirus.

Response: We observed that TMPRSS2-HiBiT signal was stabilized when putative lysine ubiquitin-acceptor sites were mutated to arginine. These data are shown here:

From this, we hypothesize that if halofuginone (HFG) proceeds through this mechanism, the mutant TMPRSS2 construct would be insensitive to HFG in terms of prevention of pseudoviral entry assay. Indeed, while HFG significantly decreased pseudoviral entry in WT TMPRSS2 cells, HFG was unable to significantly decrease pseudoviral entry in “3K-R” TMPRSS2 cells. We believe this is due to the inability of HFG activity to leverage the ubiquitination sites present on WT TMPRSS2.

5. Do TMPRSS2-K/R mutations increase signal of extracellular TMPRSS2-HiBiT?

Response: As shown above, we observed TMPRSS2 protein level to remain unaffected by HFG when key lysine residues are mutated to arginine. Further, our revision experiment demonstrated that Lysine-Mutant TMPRSS2 was insensitive to HFG-mediated decrease in pseudoviral entry. One alternative explanation for persistence of the K-R TMPRSS2 constructs in protein level, and in viral entry could be an increase in trafficking to the membrane, thus better facilitating viral entry.

To test this, we expressed WT and K-R TMPRSS2 constructs and measured non-lytic HiBiT signal using the extracellular detection kit. This allowed us to measure membrane localized TMPRSS2 for both constructs. Following this, we lysed the cell to detect total HiBiT signal. The ratio of these data represents the extra-cellular localized TMPRSS2 signal between both constructs. We observed that there was no

significant difference in extracellular:total TMPRSS2 signal between WT and K-R mutant. This suggests that at baseline, effects on viral entry are not due to increased recruitment to the membrane.

Further, we tested this system using HFG treatment. Our model suggests that the effect of HFG on TMPRSS2 proceeds in part through the degradation of TMPRSS2, thus the insensitivity of K-R TMPRSS2 mutants to HFG in terms of protein level and effect on viral entry. To further support this model, we expressed WT and K-R TMPRSS2 mutant constructs prior to treatment with HFG, and non-lytic HiBiT detection was used to measure the membrane pool of TMPRSS2 protein. We observed that HFG treatment significantly decreased WT TMPRSS2 signal, but did not have a significant effect on K-R mutant TMPRSS2.

Taken with the data above, this suggests that at baseline, the TMPRSS2 K-R mutant does not have enhanced membrane localization relative to WT, but upon HFG stress, the K-R mutant is resistant to the effect of HFG and remains membrane-localized comparable to Veh treatment. In contrast, WT TMPRSS2 protein level is decreased, and viral entry is enhanced.

6. Does HFG effect on transcription level of DCAF1?

Response: We treated Caco-2 cells with our two top screening hits, halofuginone (HFG) and homoharringtonine (HHT), prior to RNA extraction and qPCR for DCAF1 expression levels. We observed no significant change in DCAF1 expression relative to Vehicle treatment.

7. Authors found DCAF1 by siRNA screening using Beas-2b cells. In Figure 6B, many cell lines have accepted infection of SARS-CoV-2 pseudovirus. Authors should show whether siDCAF1 increases the infection in these cell lines.

Response: Our initial ubiquitination RNAi screen was conducted in airway epithelial Beas-2B cells. As noted, several cell lines have shown susceptibility to SARS-CoV-2 pseudoviral entry. We have conducted additional entry assays with DCAF1 KD in Beas-2B and in Caco-2 cells. Depletion of DCAF1 in both of these cell lines resulted in insensitivity to HFG treatment.

8. In Figure S6, proline treatment recovered the effect of HFG in vitro translation of TMPRSS2 and TMPRSS2-HiBiT signal. Can the proline treatment cancel the HFG effect for infection of SARS-CoV-2 pseudovirus in many types of cell lines?

Response: This is an intriguing question that extends our observations into the effects of proline rescuing HFG-mediated decrease in TMPRSS2 protein. We treated Caco-2 and Calu-3 cells with HFG and with co-treatment of proline, and then treated with SARS-CoV-2 pseudovirus. While HFG was able to significantly decrease pseudoviral entry in both cell lines, co-treatment with proline ablated this effect. We believe these results strengthen the underlying mechanism for HFG effect on TMPRSS2 and thus pseudoviral entry. These results are shown here and in Figure S6C and S6D.

Minor issues

1. Chemical structures of HHT and HFG help readers to understand the study.

Response: We have added the structures of HHT and HFG as Figure 2A and shown below:

2. Please explain why authors use Beas-2b cells for the screening?

Response: Beas-2b are an immortalized airway epithelial cell line which model several infectious models of lung disease, and we have optimized them for repurposed drug UPS siRNA library screening(s) (1). Beas-2B do not express ACE2, and thus they are not useful for entry assay screens. We have modified our methods to include the rationale provided above.

3. Please explain inhibition mechanism of HHT in discussion.

Response: We have included a paragraph on the mechanism of action of Homoharringtonine in the discussion. The relevant section is copied here as well:

Homoharringtonine (omacetaxine mepesuccinate) is approved for the treatment of refractory chronic myeloid leukemia in the United States². Homoharringtonine binds to free ribosomes and inhibits protein translation by preventing peptide chain elongation³. Thus, it has broad cellular effects. Its effect is most pronounced in the context of malignant cells that display increased rates of protein translation. By depleting very-short lived proteins, malignant cells treated with homoharringtonine undergo apoptotic cell death^{4, 5}. In addition to its effects on protein translation, other mechanisms of action have been explored. Notably, homoharringtonine has been shown to stimulate degradation of the fusion protein BCL-ABL⁶. Clinically, treatment with homoharringtonine is frequently accompanied by grade III or IV adverse events, including severe myelosuppression²; thus, limiting its clinical utility at high doses.

However, homoharringtonine has also been identified as an agent that can prevent SARS-CoV-2 replication *in vitro* ⁷. In particular, SARS-CoV-2 replication was diminished in homoharringtonine-treated Vero E6 cells with an EC50 estimated at 2.1 μM. Our results show that homoharringtonine acts to prevent viral entry by reducing TMPRSS2 levels, an effect that would not be observed in Vero E6 cells, since this cell line does not express TMPRSS2 (PMID: 18562527). Thus, as expected the EC50 we observed in TMPRSS2-expressing Calu-3 cells is in the nanomolar not micromolar range (Figure 8). Lastly, homoharringtonine has also been identified to have broad anti-viral activity, inhibiting replication of a number of viruses *in vitro* ^{8,9}, probably via inhibition of viral protein translation.

4. Please discuss biological roles of DCAF1 in other infection diseases.

Response: We have included additional discussion about the role of DCAF1 in other disease states. The relevant section is copied here as well:

DCAF1 is a substrate receptor of the Cullin Ring Ligase 4 (CRL4) E3 ligase complex ¹⁰. It functions to connect the CRL4 ubiquitin-transferring machinery to a number of substrate proteins that interact with various motifs within the DCAF1 protein. Published DCAF1 substrates include Merlin ¹¹, RAG1 ¹², MCM10 ¹³, and NRF2 ¹⁴. Importantly, while DCAF1 interacts with these endogenous substrates, it was initially described as a binding partner of the lentiviral protein Vpr ¹⁵. Vpr binds a number of host anti-viral proteins including SAMHD1 and UNG2, and through also interacting with DCAF1 these substrates are ubiquitinated by the CRL4 complex. This mechanism is thought to be a major contribution to viral hijacking of host protein degradation pathways. The role of DCAF1 in contributing to the pathophysiology of disease is only beginning to be understood. DCAF1 KO mice are embryonically lethal ¹⁶, and thus studies with conditional DCAF1 knockout or transient down-regulation have been used to study its function. Most recently, DCAF1 was shown to play a key role regulatory T cell senescence and ageing ¹⁷. Thus, our results add to the growing body of literature on DCAF1 and implicate it as a factor regulating stability of the host protease TMPRSS2.

Reviewer #2 (Remarks to the Author):

Yanwen Chen & al. developed an original assay to identify small molecules that inhibit the expression of TMPRSS2, a cellular protease essential for SARS-CoV-2 entry into target cells. In total, 2700 compounds were screened and 24 candidate molecules were selected for validation. The best candidates, homoharringtonine (HHT) and halofuginone (HFG), were further investigated for their ability to suppress TMPRSS2 expression and impair viral entry. Interestingly, halofuginone has been shown to modulate TMPRSS2 levels through DCAF-1 ubiquitination and proteasomal-mediated degradation, providing insights on the mode of action of this molecule. This study is well performed, original, and conclusions are mostly supported by the experimental data presented.

Comments:

- Fig. S1: Please provide the signal/background ratio for the assay. Define CXH (cycloheximide?) and provide CHX concentration used in the pilot experiment

Response: CHX is defined as cycloheximide and we used 0.1mg/mL. These data have been added to the text and figure legend.

- Where does the FDA-approved library come from? Please provide the information.

Response: The FDA-approved library was acquired from Selleck (product # L1300), and it contained 2560 compounds. This information has been added to the methods

- Normalized HiBit signals are presented in Fig. 2. Please explain the normalization procedure. How many control wells were present in each screening plate?

Response: In the course of the screen, FDA-approved compounds were stamped to 384-well plates with 64 vehicle wells per plate. To normalize the signal for both extracellular and total HiBiT signal detection, raw luminescent values were normalized to vehicle treatments within each plate and represented as log-scale to vehicle. Fig. 2 represents the two sets of normalized values plotted against each other. We have included the individual data points in the numerical data supplemental file.

- Information is missing on the toxicity of tested molecules, thus impeding the interpretation of Fig. 2. Was the NanoLuc signal inhibited because TMPRSS2 is selectively suppressed or because cells were overly stressed and dying. For example, Halofuginone is highly toxic at 10 microM as shown in Fig. 3. Although toxicity assays were performed on the selected molecules in a second time, which is fine, this is a limitation of the initial assay which should be clearly stated in the text.

Response: We have added language about toxicity as a limiting factor in the assay design, which prompted the dose course IC50 determination and measurement of cell viability.

- It is said that in total 24 compounds were selected from the screen and retested (see the main text). Dose-response curves and validation experiments are presented for only 16 compounds (6 in Fig. 2 and 10 in Fig. S2). Please explain. Were the other compounds toxic or failed to retest?

Response: We have included a new schematic detailing our screening protocol and selection of compounds. It is shown in Fig. 2A and here:

Briefly, we observed 100 compounds that decreased the surface TMPRSS2 level (as detected by extracellular HiBiT assay) by >60%. From these, we investigated the list of hit compounds and distilled down to 24 compounds that were the top clinically relevant compounds. These 24 compounds were validated by IC₅₀ determination – these data are included in the figures and supplemental figures. Of the 24 top hits, the two most potent compounds were halofuginone and homoharringtonine.

- Fig. 3 and S2. Inhibition curves with IC₅₀s are presented. Were the observed inhibitions statistically significant? P-values must be provided.

Response: To test this, we conducted a *t*-test between the highest dose and the vehicle treatment for each of the top 24 hits. These are represented below:

Compound	P-Value t -test (Veh vs 10μM)	
	Extracellular HiBiT	Lytic HiBiT
Afatinib (BIBW2992)	0.002474	0.003780
Auranofin	0.000684	0.000008
Azelnidipine	0.000127	0.000001
Cilnidipine	0.000134	0.000042
Costunolide	0.002162	0.000941
Dasatinib	0.000189	0.001489
Dasatinib hydrochloride	0.002034	0.001935
Disodium Phosphate	0.380713	0.593085
Docetaxel	0.029062	0.224661
Efonidipine	0.117115	0.004369
Flubendazole	0.181234	0.033963
Gefitinib (ZD1839)	0.453778	0.002866
Halofuginone	0.000185	0.000184
Homoharringtonine	0.000023	0.000077
Isradipine	0.000084	0.007047
Phenazine methosulfate	0.000077	0.005363
pyrvinium	0.000002	0.000021
Sanguinarine	0.002116	0.000809
Sanguinarine chloride	0.002920	0.001034
Tanshinone IIA	0.151080	0.749249
Triamcinolone Acetonide	0.970733	0.617204

Venetoclax (ABT-199, GDC-0199)	0.000142	0.000012
Verteporfin	0.000002	0.000107
Zileuton	0.131600	0.707399

- Fig S2. The scale of the graph showing Costunolide effect on Lytic HiBit signal needs to be corrected and should range from 0-150%.

Response: We have modified this data representation to have the same range for both extracellular and lytic HiBiT signals. This is also shown here:

- "These agents were also directly assessed by Western blot analysis for TMPRSS2-HiBiT protein expression (fig. S3)." This sentence should be rephrased because only homoharringtonine, halofuginone and venetoclax were analyzed by WB.

Response: We have rephrased this comment to reflect that these top screening hits were analyzed by immunoblotting.

- Fig. 4. Homoharringtonine or halofuginone reduce TMPRSS2 but not actin or LacZ-V5 expression in lung cells. Other membrane proteins such as E-cadherin (as displayed in Fig. 5f for HFN) should be presented to validate the relative specificity of tested compounds for TMPRSS2

Response: We have analyzed lysate from Caco-2 cells treated with HFG or HHT dose course and measured e-cadherin protein levels by immunoblotting. We observe no compound effect on e-cadherin in contrast to the compound effect on TMPRSS2. These data are included in Fig. S5 and are repeated here as well.

- In line with the previous comment, ACE2 expression in the presence of HHT or HFN should be analyzed. This is essential to validate conclusions from Fig.6.

Response: This is a valid concern for the proper interpretation of our screening data. We have expressed ACE2-GFP with TMPRSS2-V5 and treated with HFG or HHT prior to immunoblotting. Here we observe a strong susceptibility of TMPRSS2 protein loss, but Ace2 protein remains unaffected by compound treatment. These data are shown here and in Fig. S5. From these results, we believe that our screening hits are not working through effect on ACE2.

- Fig. 5D. A plasmid encoding TMPRSS2-HIS-V5 is transfected, but TMPRSS2-V5 expression is indicated on the WB. This needs to be corrected.

Response: We have expanded the label for the TMPRSS2-V5-HIS on the immunoblot data as requested; however, we used the shorter label for both conciseness and clarity – the TMPRSS2 signal was detected through immunoblotting of the V5 epitope, not the HIS-based epitope. We defer to the reviewer suggestions.

Further we have labeled as “TMPRSS2-V5-HIS”, instead of your suggested “TMPRSS2-HIS-V5” as the HIS epitope is located downstream of the V5.

- Fig. S5. Please provide cellular viability for Calu-3 cells when treated with HHT or HFG.

Response: We have included loading controls of Calu-3 lysates analyzed through immunoblotting and corrected protein densitometry data to these controls to reflect any cellular toxicity. Further, our SARS-CoV-2 pseudoviral entry assays with Calu-3 are corrected to total cellular viability using Cell-TiterGlo2.0 measurements following HiBiT entry assays. Data represented are corrected to these viability measurements before any normalization to controls. We have updated our figure labeling and legend statements to reflect these important steps.

- Fig. 6A. The binding of viral particles tagged with the HiBiT peptide is expected to generate background signal when using the lytic reagent. This point needs to be addressed. Explain how viral binding to cellular membrane is distinguished from actual viral entry.

Response: The course of this assay is based on the idea that HiBiT-tagged SARS-CoV-2 proteins assembled into pseudoviral particles will enter infected cells and then form functional luciferase enzymes upon lysis, with luminescence acting as a surrogate measure for overall viral entry. As shown in our manuscript we have prepared this figure to illustrate the concept:

The protocol of detection of infected cells involves extensive washing of cells prior to their lysis, concentration, and re-constitution of the nanoluciferase enzyme. Specifically, at least 5 washes with PBS are used, which will be enough to remove any non-entered viral particle. We have included this information in the methods section. Also, the data observed using this quick-HiBiT pseudoviral entry system is consistent in our observed IC50 values compared to those determined with decrease in TMPRSS2 protein via HiBiT assay and immunoblotting. This consistency gives us confidence in the robustness of this assay.

The drug potency measured in this FLUC luciferase based pseudoviral entry assay is highly similar to our HiBiT pseudoviral entry experiment, which validates our assay.

- Fig. 6c and d. Please indicate that presented experiments were conducted in Caco-2 cells.

Response: We have included labels to show these experiments were run with Caco-2 cells.

- In their conclusion, authors should clearly indicate that in addition to TMPRSS2, HHT and HFN potentially impair the expression of other membrane factors involved in viral entry. This cannot be excluded at this point.

Response: This is a valid point for which we have added some discussion to in the manuscript text. Specifically, in this screen we looked directly at compounds that modulate TMPRSS2 protein level. We cannot rule out that these compounds may affect other family member proteins or membrane factors involved in viral entry. Ideally, these hit compounds would be tested for differential proteomics to uncover significantly changing proteins upon treatment, however we believe such an experiment is beyond the scope of this manuscript. Importantly, the above experiments demonstrate that the two top hits do not have a noticeable effect on e-cadherin or Ace2 protein level, suggesting a level of selectiveness in their effect.

Reviewer #3 (Remarks to the Author):

In this manuscript, Chen et al. use a repositioning screen of a library of FDA-approved or clinically tested drugs to identify compounds that downregulate TMPRSS2 expression, and thereby limit SARS-CoV-2 entry. The focus of this study is of great interest in the current pandemic as effective antivirals against SARS-CoV-2 remain an unmet clinical need. The repurposing of clinically tested drugs could allow a rapid transition to an in vivo evaluation, and the focus on TMPRSS2 expression uses a different approach than similar screens.

The study identifies two lead molecules and proposes a new mechanism of action of halofuginone, the most promising compound, based on DCAF-1-mediated proteasomal degradation.

To my knowledge, antiviral activity of halofuginone against SARS-CoV-2 has not previously been described. Moreover, the insights into a potential mechanism of action for the antiviral activity of this compound are novel. The study is generally well controlled and statistical analyses are applied appropriately. However, I do think that the description of individual experiments could be improved by providing additional details. In cases where the authors refer to previous publications, providing a brief summary of the experimental approach would be helpful for the reader.

While I support the publication of this study in general, I have the following suggestions for the authors to strengthen the manuscript:

A current weakness of the paper is the lack of detailed information on both the setup/analysis and the results of the compound screen.

1) The results of the screen should be made available in their entirety. Drug repurposing datasets, such as the one generated in this study, are of great value for drug discovery efforts and secondary analyses that are particularly relevant in light of the current pandemic. Unless there are compelling reasons precluding publication of the complete dataset, it should be made available as supplementary information or through a public data repository.

Response: We have expanded the detail and methods of the drug screening data in the protocol section and in the results. We have added a new graphical figure to explain the workflow, also shown here:

We have including the individual data from the screening process in the supplemental numerical data file accompanying the manuscript submission.

2) Additional details describing the setup and data analysis of the screen should be provided. What positive and negative controls were used (Fig S1 shows “Veh” and “CHX”, but no explanation is provided)? How was the data normalized? Were replicates included in the screen, and if so, what was the correlation?

Response: We have expanded the methods section regarding the screening design, implementation, and analysis. Specifically, in FigS1, we prepared 64 well replicates of TMPRSS2-HiBiT expressing Beas-2B cells in a 384-well format. These wells were either treated with vehicle or cycloheximide (0.1mg/mL for 6hr) prior to collection and Lytic HiBiT luminescent detection. The plot (also shown below) indicates the spread of detected luminescent signal among replicates for each treatment group. As we were interested in screening compounds that decreased TMPRSS2 protein level (as measured by HiBiT signal), we used cycloheximide as a positive control for decrease in TMPRSS2. As cycloheximide is an inhibitor of protein synthesis and an often-used compound for measuring protein stability, its use would represent a lower end to the theoretical assay signal window. From these data, we calculated a Z'-factor value for the assay (Zhang et al., 1999); we used this value to represent overall reproducibility and signal window for TMPRSS2-HiBiT as a platform for screening.

3) Showing the controls (positive and negative) in Figures 2A-C would help the reader to assess the impact of the identified compounds.

Response: We have now included labeling of the vehicle treatment on each of the plots to better orient the reader to the results. They now appear as this in figure S2B-C:

4) The manuscript currently does not provide a clear narrative how 24 compounds were selected out of the ~100 hits, and which ones could be validated. A supplementary table/file showing the screening results should be provided for all ~100 identified compounds and indicate which compounds were selected for follow up; it is currently unclear which 24 compounds were chosen and why.

Response: We have expanded the description and protocol of the screening process. We have also added a graphical figure to show the workflow of the screening assays, also shown here:

We first thresholded the screening results by those that reduced surface TMPRSS2 protein level (via Extracellular HiBiT Assay) by >60% - this yielded 100 compounds. We examined these hits and removed clinically non-relevant compounds from consideration, for example - Arsenic oxide was detected to have an effect in reducing TMPRSS2, however this compound, while FDA-approved as a chemotherapeutic, will likely not be clinically appropriate for patients with Covid19. From this we arrived at 24 compounds, which were validated by IC50 determination. From this, halofuginone and homoharringtonine demonstrated the highest potency and efficacy.

5) Only 16 of 24 selected compounds are shown in dose response (Fig 3, S2) and 3 by western blot (Fig S3). This data needs to be shown for all 24 compounds and indicate which compounds are considered active. This will also allow an assessment of the screen reconfirmation rate.

Response: Related to the response to point 4, we have improved the description of the assay process. We have included the dose-course data of all 24 clinically relevant hits recovered from the screening process. Further, the raw data from the earlier steps will be included as a supplementary spreadsheet file with the manuscript.

The dose response curves in Fig 3 and S2 do not seem to fit the data in all cases. Particularly for Dasatinib and Docetaxel the respective fit is poor and unlikely to provide a correct IC50. In other cases, including Homoharringtonine and Halofuginone, the curve fit suggests a stronger maximal effect than the measured dose response data supports, which may affect the calculated IC50.

Response: We have re-calculated and represented the nonlinear fit for these compounds. Our original nonlinear fits for Dasatinib were conducted using a standard slope, equivalent to a Hill slope of -1.0. For Dasatinib our original fit appears as follows:

To better represent the data pattern, we used a Sigmoidal fit, resulting in the following fit:

These models better fit the pattern of data, and the re-calculated IC₅₀ values represent the concentration for 50% loss in TMPRSS2 signal. However, it is important that we examine the absolute decrease of the data/model with dose, as the maximal decrease in TMPRSS2 by Dasatinib appears to be just below 50%. In contrast, one of our top hits Halofuginone shows ability to decrease TMPRSS2 signal much further:

6) How were the IC₅₀ values calculated, and could a model with additional parameters (e.g. a four-parameter logistic equation) provide a better fit?

Response: We used Graphpad PRISM 9.0 to prepare these models. Related to the above point of poor fit, we have re-calculated the models using the Sigmoidal fit function. This results in a much better model fit, and with IC₅₀ values representing compound concentration for absolute 50% decrease. As shown above, the model for Dasatinib has changed from original (left) to revised (right):

7) The text indicates that the drugs' toxicity (CC₅₀) and SI were determined. These values should be provided whenever a CC₅₀ could be determined.

Response: We have included these values in our results using the PD/PK data available. These data are also included in our table 1, shown below as well.

Table 1

Drug	Disease	Regulatory Status	Dose	Route (frequency)	Cmax	In vitro IC50 (viral entry)	Reference
Homoharringtonine (Omacetaxine)	chronic myeloid leukemia (CML)	FDA approved	1.25 mg/m ²	SC (BID)	55nM	~30nM	[32]
Halofuginone	scleroderma	Phase 1/2	3.5mg/day	Oral	7nM	~30nM	[33]
Verteporfin	photosensitizer for photodynamic therapy	FDA approved	0.3mg/kg	IV (within 45min)	1.92µM	~10µM	FDA
Cilnidipine	Hypertension	FDA approved	10mg	Oral (QD)	18.1nM	~3µM	[34]
Dasatinib	chronic myelogenous leukemia (CML) and acute lymphoblastic leukemia (ALL)	FDA approved	140mg	Oral (QD)	0.307µM	>10µM	[35]
Venetoclax	chronic lymphocytic leukemia (CLL) or small lymphocytic lymphoma (SLL)	FDA approved	400mg	Oral (QD)	1.27µM	>10µM	[36]

The results section describes that measured IC50 values were compared to published PK properties, which I assume is referring to Table 1.

8) If so, I recommend referring to Table 1 in the results section and clearly state from which experiment the listed IC50 values are derived.

Response: We have added a more direct callout to Table 1 in our result section, and have clearly indicated the source experiment for the reported ic50 (viral entry assay)

Table 1

Drug	Disease	Regulatory Status	Dose	Route (frequency)	Cmax	In vitro IC50 (viral entry)	Reference
Homoharringtonine (Omacetaxine)	chronic myeloid leukemia (CML)	FDA approved	1.25 mg/m ²	SC (BID)	55nM	~30nM	[32]
Halofuginone	scleroderma	Phase 1/2	3.5mg/day	Oral	7nM	~30nM	[33]
Verteporfin	photosensitizer for photodynamic therapy	FDA approved	0.3mg/kg	IV (within 45min)	1.92µM	~10µM	FDA
Cilnidipine	Hypertension	FDA approved	10mg	Oral (QD)	18.1nM	~3µM	[34]
Dasatinib	chronic myelogenous leukemia (CML) and acute lymphoblastic leukemia (ALL)	FDA approved	140mg	Oral (QD)	0.307µM	>10µM	[35]
Venetoclax	chronic lymphocytic leukemia (CLL) or small lymphocytic lymphoma (SLL)	FDA approved	400mg	Oral (QD)	1.27µM	>10µM	[36]

9) I was unable to verify the PK data provided in Table 1 since the references appear to be incorrect or mixed up; please update the references.

Response: We have updated the references in Table 1 to correctly reflect the cited PK data studies.

An important caveat of the manuscript is the absence of infection assays with live SARS-CoV-2 that confirm the activity of the compounds in the context of viral infection and replication. It should be noted that the activity of homoharringtonine against SARS-CoV-2 has been investigated in a recent peer-reviewed study. Choy et al. (<https://doi.org/10.1016/j.antiviral.2020.104786>) report an EC50 for the inhibition of SARS-CoV-2 by homoharringtonine in Vero E6 cells of 2.1uM, supporting this compound's activity. Nevertheless, and despite the differences in experimental systems, the data raises the question to what extent the results of the pseudovirus entry assay conducted in this study (~30nM) are predictive of the activity against SARS-CoV-2 viral infection.

10) Since an infection with live SARS-CoV-2 could not be performed, it would be helpful to discuss these potential caveats. This is especially relevant given the narrow predicted therapeutic window for the lead compounds when comparing the IC50 data to available PK properties.

Response: This is an important point raised by the reviewer. To address this point, and those raised by other reviewers and the editor, we conducted live viral infectivity assays with SARS-CoV-2 with treatment of our two top screening hits, HFG and HHT.

We worked with the CRO IITRI (Chicago, USA) to test a dose course of halofuginone and homoharringtonine in Calu-3 human lung cells with infection of SARS-CoV-2 virus. This protocol utilized the viral entry assay as developed by the Pöhlmann lab¹. Briefly, human Calu-3 lung cells were pre-treated with the hit compounds for 24 hours prior to viral entry assay. Treated cells were washed and inoculated with two doses (0.01 MOI or 0.001 MOI) of SARS-CoV-2 (wildtype-USA-WA1/2020) for 75 min. After exposure period, cells were washed and fresh media containing homoharringtonine or halofuginone was added. Following 48 and 72hrs, cell supernatant was collected for qPCR analysis of viral transcripts. We also represent this workflow graphically here and in Figure 8A.

We observed that both HHT and HFG had a strong dose-dependent effect in blunting SARS-CoV-2 viral infection at both viral doses and time points.

48hr, MOI 0.01

48hr, MOI 0.001

72hr, MOI 0.01

72hr, MOI 0.001

Of note, these compounds demonstrated efficacy in blunting SARS-CoV-2 at doses comparable to those we observed of cell-based pseudoviral entry assay and of TMPRSS2 protein reduction assays. From these data, we show functional relevance of our screening hits with live-wild-type SARS-CoV-2 assays.

Regarding the work of Choy et al. in regard to homoharringtonine inhibiting SARS-CoV-2 in VeroE6 cells, we believe the discrepancy in efficacy may come from the levels of TMPRSS2 protein present in VeroE6 compared to higher-expressing TMPRSS2 cell lines such as Caco-2.

The authors propose a mechanism of action for halofuginone based on DCAF-1-mediated proteasomal degradation of TMPRSS2. In addition, they show evidence for a second, previously described MOA based on the inhibition of prolyl-tRNA synthetase. However, the contribution of these mechanisms to the downregulation of TMPRSS2 by halofuginone remains unclear and further insight into their biological relevance would be of great interest.

11) The data shown in Fig S6B seems to indicate that the addition of proline completely abolishes the effect of HFG on TMPRSS2 expression. If rescuing prolyl-tRNA synthetase inhibition eliminates the effect of HFG on TMPRSS2 levels, what is the relevance of HFG-mediated proteasomal degradation of TMPRSS2? Could the inhibition of pseudoviral entry be reversed by the addition of proline?

Response: This is a good point that was similarly raised by another reviewer. To address the role of proline in the effect of HFG-dependent inhibition of pseudoviral entry, we treated Caco-2 and Calu-3 cells with HFG and with co-treatment of proline, and then treated with SARS-CoV-2 pseudovirus. While HFG was able to significantly decrease pseudoviral entry in both cell lines, co-treatment with proline ablated this effect. We believe these results strengthen the underlying mechanism for HFG effect on TMPRSS2 and thus pseudoviral entry. These results are shown here and in Figure S6C-D:

We also believe that proteasomal degradation plays a role in the efficacy of halofuginone, as depletion of DCAF1 in both of these cell lines resulted in insensitivity to HFG treatment in both Caco-2 and Beas-2B cell lines.

From this, we see two potential avenues by which HFG is modulating TMPRSS2 protein level, and thus cellular susceptibility to viral entry.

The data in Figure S10 appears too preliminary for publication and should be revised or removed.

12) The experiment needs to be conducted in replicates, including a statistical evaluation.

Response: We have revised these data by providing additional replicates, allowing for better fit. We now have $n=3$ per temperature and sample. These data now appear as follows:

Additional minor points:

13) What is the rationale for the selection of compounds highlighted in Fig 2C, since they do not seem to match those listed in Fig 2A/B?

Response: We have added additional information about the workflow process. As shown above, we have a diagram added to the figures to better explain the screening workflow.

We first thresholded the screening results by those that reduced surface TMPRSS2 protein level (via Extracellular HiBiT Assay) by $>60\%$ - this yielded 100 compounds. We examined these hits and removed clinically non-relevant compounds from consideration, for example - Arsenic oxide was detected to have an effect in reducing TMPRSS2, however this compound, while FDA-approved as a chemotherapeutic, will likely not be clinically appropriate for patients with Covid19. From this we arrived at 24 compounds, which were validated by IC_{50} determination. From this, halofuginone and homoharringtonine demonstrated the highest potency and efficacy.

14) In Fig 3, please define what the dotted lines along the modeled dose response curve represent.

Response: The dotted lines represent the 95% confidence interval for the model. We have added this information to the figure legend.

15) Fig 4A: What compound concentrations were used?

Response: For this experiment we used HFG at 3 μ M in combination with carfilzomib (CFZ, 1 μ M) or bafilomycin A1 (BafA1, 1 μ M). This information has been added to the figure legend.

16) Fig 4B: How was the screen normalized and what controls were used? A brief summary of the experimental setup in addition to referencing the original publication would be helpful.

Response: We have added information about the processing and normalization of the ubiquitination siRNA screening process to the methods section.

17) The label for Fig 4C is missing.

Response: We have noted this and added to the current revision figure.

18) Fig 4F: What timepoint does the data represent?

Response: This experiment was conducted with 6 hours of HFG treatment

19) Fig 6H: A (representative) western blot showing the knockdown level of DCAF1 should be included.

Response: We have added a representative blot for DCAF1 KD studies. This is shown in Fig 7H and

below:

20) Statistics need to be added to Fig S6B, as well as a brief description of the experimental setup.

Response: We have added statistical testing to this assay, it is now shown here:

Further description of the experimental set up has been added to the methods section.

References:

1. Hoffmann M, *et al.* SARS-CoV-2 Cell Entry Depends on ACE2 and TMPRSS2 and Is Blocked by a Clinically Proven Protease Inhibitor. *Cell*, (2020).
2. Alvandi F, *et al.* U.S. Food and Drug Administration approval summary: omacetaxine mepesuccinate as treatment for chronic myeloid leukemia. *Oncologist* **19**, 94-99 (2014).
3. Gandhi V, Plunkett W, Cortes JE. Omacetaxine: a protein translation inhibitor for treatment of chronic myelogenous leukemia. *Clin Cancer Res* **20**, 1735-1740 (2014).
4. Chen J, *et al.* Homoharringtonine targets Smad3 and TGF- β pathway to inhibit the proliferation of acute myeloid leukemia cells. *Oncotarget* **8**, 40318-40326 (2017).
5. Sun Q, *et al.* Homoharringtonine regulates the alternative splicing of Bcl-x and caspase 9 through a protein phosphatase 1-dependent mechanism. *BMC Complement Altern Med* **18**, 164 (2018).
6. Li S, Bo Z, Jiang Y, Song X, Wang C, Tong Y. Homoharringtonine promotes BCR \rightarrow ABL degradation through the p62 \rightarrow mediated autophagy pathway. *Oncol Rep* **43**, 113-120 (2020).
7. Choy K-T, *et al.* Remdesivir, lopinavir, emetine, and homoharringtonine inhibit SARS-CoV-2 replication in vitro. *Antiviral research* **178**, 104786-104786 (2020).
8. Andersen PI, *et al.* Novel Antiviral Activities of Obatoclox, Emetine, Niclosamide, Brequinar, and Homoharringtonine. *Viruses* **11**, (2019).
9. Dong HJ, *et al.* The Natural Compound Homoharringtonine Presents Broad Antiviral Activity In Vitro and In Vivo. *Viruses* **10**, (2018).
10. Nakagawa T, Mondal K, Swanson PC. VprBP (DCAF1): a promiscuous substrate recognition subunit that incorporates into both RING-family CRL4 and HECT-family EDD/UBR5 E3 ubiquitin ligases. *BMC Mol Biol* **14**, 22 (2013).
11. Li W, *et al.* Merlin/NF2 suppresses tumorigenesis by inhibiting the E3 ubiquitin ligase CRL4(DCAF1) in the nucleus. *Cell* **140**, 477-490 (2010).
12. Schabla NM, Perry GA, Palmer VL, Swanson PC. VprBP (DCAF1) Regulates RAG1 Expression Independently of Dicer by Mediating RAG1 Degradation. *J Immunol* **201**, 930-939 (2018).
13. Chang H, *et al.* Distinct MCM10 Proteasomal Degradation Profiles by Primate Lentiviruses Vpr Proteins. *Viruses* **12**, (2020).
14. Chen Y, *et al.* A small molecule NRF2 activator BC-1901S ameliorates inflammation through DCAF1/NRF2 axis. *Redox Biol* **32**, 101485-101485 (2020).
15. Ahn J, Vu T, Novince Z, Guerrero-Santoro J, Rapic-Otrin V, Gronenborn AM. HIV-1 Vpr loads uracil DNA glycosylase-2 onto DCAF1, a substrate recognition subunit of a cullin 4A-ring E3 ubiquitin ligase for proteasome-dependent degradation. *J Biol Chem* **285**, 37333-37341 (2010).
16. McCall CM, *et al.* Human immunodeficiency virus type 1 Vpr-binding protein VprBP, a WD40 protein associated with the DDB1-CUL4 E3 ubiquitin ligase, is essential for DNA replication and embryonic development. *Mol Cell Biol* **28**, 5621-5633 (2008).
17. Guo Z, *et al.* DCAF1 regulates Treg senescence via the ROS axis during immunological aging. *J Clin Invest* **130**, 5893-5908 (2020).

Reviewers' Comments:

Reviewer #3:

Remarks to the Author:

Authors have well addressed my comments.

Few minor points and typos should be corrected in the final version:

- Table I: Please replace "Fig. XX" by the appropriate figure.
- Fig S5a, S5b, S6c, S6d S7f and S7g are interesting. However, they are not presented and discussed in the revised manuscript.

Reviewer #4:

Remarks to the Author:

The revised manuscript "A high throughput screen for TMPRSS2 expression identifies FDA-approved and clinically advanced compounds that can limit SARS-CoV-2 entry" from Chen et al. addresses many of the original concerns and strengthens the conclusions. In particular, the newly added data showing an effect on infection with replication competent SARS-CoV-2 is valuable to confirm the findings and support the relevance of the study.

A remaining concern is the availability of the screening data. I had recommended that the results of the primary screen should be made publicly available. Especially the names and activities of the 100 primary hits need to be included in the manuscript to allow an evaluation of the screen that goes beyond the 24 compounds that were somewhat arbitrarily selected for follow up. While a supplementary data table containing this information has been provided for review, this data table is not at all mentioned in the manuscript and it is unclear whether the table is intended for publication. I believe it is critical that this screen data is included in the published manuscript as supplemental information, and it should be referred to in the text.

Moreover, it appears that some of the changes described in the rebuttal letter were not actually implemented in the revised manuscript. These issues, listed below, can be addressed without conducting further experiments, but should be corrected prior to publication of the manuscript.

The authors state that the methods section has been expanded with further details on screening design, implementation, and analysis. However, the respective methods section of the manuscript remains virtually unchanged. A description of the screen normalization is still missing, and the negative control (vehicle) is not defined (DMSO?). The drug library (L1300) should also be specified in the methods section (it is currently only shown in Fig 2A.)

In Figures 3 and S2 the CC50 values of the compounds' toxicities should be calculated and shown to allow assessment of the potential therapeutic index, as described in the manuscript text. The rebuttal states that these data have been included in table 1, however this is not the case. The CC50 values should be calculated and provided in Fig 3 and S2, as done for IC50 values.

The authors state that they have "clearly indicated the source experiment for the reported IC50" in Table 1, however the table only lists "Fig. XX" for each compound. Please correct this oversight and provide the necessary information.

Moreover, the publication references listed in Table 1 are still incorrect and do not correspond to the respective compounds.

Please check if panels c, d, and e of Fig.7 are correctly referenced in the text. It appears that panels c and e are mixed up.

Please name what cell line was used in the dose response assays shown in Fig. 3 and S2 in the figure legends.

The rebuttal states that the experiment now shown in Fig. 6A was conducted after 6 hours of HGF treatment. This timepoint should be stated in the figure legend.

REVIEWERS' COMMENTS

Please note that we were unable to obtain a report from reviewer 1 and asked reviewer 2 to comment on your response to their concerns. Please also address the following comments from reviewer 2 in a revised manuscript:

Reviewer #1/2:

- Authors should make sure that all figures and supplementary figures are discussed in the text. Several supplementary figures that were added to address reviewers' comments are not presented in the results or discussion section. This is the case for Fig. S5a, S5b, S6c, S6d, S7f and S7g. In addition, panels C, D and E in Fig. 7 should be relabelled E, C and D, respectively.

Response: We thank Reviewer 2 for filling in for Reviewer 1. We have added call-outs to these new panels and corrected the ordering for Figure 7.

- In their answer to Reviewer #1, authors say that « Beas-2B do not express ACE2, and thus they are not useful for entry assay screens. » (see the answers to Reviewer #1, minor point 2). It is thus perfectly fine to use Beas-2B when studying TMPRSS2 expression and I agree with that. However, it seems that Beas-2B were also used as target cells to monitor the entry of pseudotyped SARS-CoV-2 particles in panels Fig. 7i and Fig S7f. This could be a labeling error because the legend of Fig. 7i refers to Caco-2, not Beas-2B. However, the legend of S7f mentions Beas-2B. This needs to be clarified.

Response: Thank you for pointing out the labeling error in Fig. 7i, we have corrected the figure labeling to match the figure legend as these are indeed Caco-2 cells assays. Regarding the supplemental entry assay (currently in Fig. S11C), these cells are correctly labeled as Beas-2B cells in an attempt to address the reviewer concerns that entry assay data be conducted in 'many different cell lines'. While Beas-2B indeed show lower entry ability relative to other cell lines (please see Fig 7B) they still have capacity for viral entry, which is augmented with OE of TMPRSS2. For this reason, we proceeded with Caco2 as our primary cell line for DCAF1 KD and entry assay. Further, we were able to get effective KD with Calu-3 cells, leading us to choose Beas-2B cells as validation to our original Caco-2 observation.

Reviewer #2 (Remarks to the Author):

Authors have well addressed my comments.

Few minor points and typos should be corrected in the final version:

- Table I: Please replace "Fig. XX" by the appropriate figure.

Response: We have included the correct callouts for the entry assay data found in Table 1. The manuscript has also been closely inspected for additional editorial errors or oversights.

- Fig S5a, S5b, S6c, S6d S7f and S7g are interesting. However, they are not presented and discussed in the revised manuscript.

Response: We have discussed these panels and included new callouts in the manuscript text.

Reviewer #3 (Remarks to the Author):

The revised manuscript "A high throughput screen for TMPRSS2 expression identifies FDA-approved and clinically advanced compounds that can limit SARS-CoV-2 entry" from Chen et al. addresses many of the

original concerns and strengthens the conclusions. In particular, the newly added data showing an effect on infection with replication competent SARS-CoV-2 is valuable to confirm the findings and support the relevance of the study.

A remaining concern is the availability of the screening data. I had recommended that the results of the primary screen should be made publicly available. Especially the names and activities of the 100 primary hits need to

be included in the manuscript to allow an evaluation of the screen that goes beyond the 24 compounds that were somewhat arbitrarily selected for follow up. While a supplementary data table containing this information has been provided for review, this data table is not at all mentioned in the manuscript and it is unclear whether the table is intended for publication. I believe it is critical that this screen data is included in the published manuscript as supplemental information, and it should be referred to in the text.

Response: We apologize for the confusion as we are including the screening data as part of the 'source data' supplemental file which also includes numerical data for all plots shown. These screening results will be made publicly available as part of the manuscript's supplemental material. We have added a callout in the text for the reader to refer to the source data file.

Moreover, it appears that some of the changes described in the rebuttal letter were not actually implemented in the revised manuscript. These issues, listed below, can be addressed without conducting further experiments, but should be corrected prior to publication of the manuscript.

The authors state that the methods section has been expanded with further details on screening design, implementation, and analysis. However, the respective methods section of the manuscript remains virtually unchanged. A description of the screen normalization is still missing, and the negative control (vehicle) is not defined (DMSO?). The drug library (L1300) should also be specified in the methods section (it is currently only shown in Fig 2A.)

Response: We have included additional language about the nature of the screening, including the catalog # of the library, the vehicle controls present, the normalization process, and analysis. We hope this better explains the screening workflow.

In Figures 3 and S2 the CC50 values of the compounds' toxicities should be calculated and shown to allow assessment of the potential therapeutic index, as described in the manuscript text. The rebuttal states that these data have been included in table 1, however this is not the case. The CC50 values should be calculated and provided in Fig 3 and S2, as done for IC50 values.

Response: We have taken the cell viability measurements in these figures and estimated a CC50 for each compound as suggested. These values are noted above the plots, similar to the IC50 determinations. Figure legends have been updated to note the CC50 values in the figure.

The authors state that they have "clearly indicated the source experiment for the reported IC50" in Table 1, however the table only lists "Fig. XX" for each compound. Please correct this oversight and provide the necessary information.

Moreover, the publication references listed in Table 1 are still incorrect and do not correspond to the respective compounds.

Response: We apologize for this oversight and have corrected the callout for the source experiments mentioned for each compound in Table 1 and have corrected the references for their clinical parameters.

Please check if panels c, d, and e of Fig.7 are correctly referenced in the text. It appears that panels c and e are mixed up.

Response: We have corrected the ordering for these panels in Figure 7.

Please name what cell line was used in the dose response assays shown in Fig. 3 and S2 in the figure legends.

Response: These dose response assays were conducted in TMPRSS2-HiBiT-BEAS-2B cells, the same as used for the original screen. We have added this detail in the figure legends for both Figure 3 and Fig. S2.

The rebuttal states that the experiment now shown in Fig. 6A was conducted after 6 hours of HGF treatment. This timepoint should be stated in the figure legend.

Response: We have added this detail in the figure legend.